# Achieving the Asymptotically Optimal Sample Complexity of Offline Reinforcement Learning: A DRO-Based Approach

**Yue Wang** *yue.wang@ucf.edu*
*Department of Electrical and Computer Engineering*
*University of Central Florida*

**Jinjun Xiong** *jinjun@buffalo.edu*
*Department of Computer Science and Engineering*
*University at Buffalo*

**Shaofeng Zou** *szou3@buffalo.edu*
*Department of Electrical Engineering*
*University at Buffalo*

**Reviewed on OpenReview:** *https://openreview.net/forum?id=Y7FbGcjOuD*

## Abstract

Offline reinforcement learning aims to learn from pre-collected datasets without active exploration. This problem faces significant challenges, including limited data availability and distributional shifts. Existing approaches adopt a pessimistic stance towards uncertainty by penalizing rewards of under-explored state-action pairs to estimate value functions conservatively. In this paper, we show that the distributionally robust optimization (DRO) based approach can also address these challenges and is asymptotically minimax optimal. Specifically, we directly model the uncertainty in the transition kernel and construct an uncertainty set of statistically plausible transition kernels. We then show that the policy that optimizes the worst-case performance over this uncertainty set has a near-optimal performance in the underlying problem. We first design a metric-based distribution-based uncertainty set such that with high probability the true transition kernel is in this set. We prove that to achieve a sub-optimality gap of $\epsilon$, the sample complexity is $\mathcal{O}(S^2 C^{\pi^*} \epsilon^{-2}(1-\gamma)^{-4})$, where $\gamma$ is the discount factor, $S$ is the number of states, and $C^{\pi^*}$ is the single-policy clipped concentrability coefficient which quantifies the distribution shift. To achieve the optimal sample complexity, we further propose a less conservative value-function-based uncertainty set, which, however, does not necessarily include the true transition kernel. We show that an improved sample complexity of $\mathcal{O}(S C^{\pi^*} \epsilon^{-2}(1-\gamma)^{-3})$ can be obtained, which asymptotically matches with the minimax lower bound for offline reinforcement learning, and thus is asymptotically minimax optimal.

## 1 Introduction

Reinforcement learning (RL) has achieved impressive empirical success in many domains, e.g., (Mnih et al., 2015; Silver et al., 2016). Nonetheless, most of the success stories rely on the premise that the agent can actively explore the environment and receive feedback to further promote policy improvement. This trial-and-error procedure can be costly, unsafe, or even prohibitory in many real-world applications, e.g., autonomous driving (Kiran et al., 2021) and health care (Yu et al., 2021a). To address the challenge, offline (or batch) reinforcement learning (Lange et al., 2012; Levine et al., 2020) was developed, which aims to learn a competing policy from a pre-collected dataset without access to online exploration.

A straightforward idea for offline RL is to use the pre-collected dataset to learn an estimated model of the environment, and then learn an optimal policy for this model. This approach performs well when the dataset

sufficiently explored the environment, e.g., (Agarwal et al., 2020a). However, under more general offline settings, the static dataset can be limited, which results in the distribution shift challenge and inaccurate model estimation (Kidambi et al., 2020; Ross & Bagnell, 2012; Li et al., 2022). Namely, the pre-collected dataset is often restricted to a small subset of state-action pairs, and the behavior policy used to collect the dataset induces a state-action visitation distribution that is different from the one induced by the optimal policy. This distribution shift and the limited amount of data lead to uncertainty in the estimation of the model, i.e., transition kernel and/or reward function.

To address the above challenge, one natural approach is to first quantify the uncertainty, and further take a pessimistic (conservative) approach in face of such uncertainty. Despite of the fact that the uncertainty exists in the transition kernel estimate, existing studies mostly take the approach to penalize the reward function for less-visited state-action pairs to obtain a pessimistic estimation of the value function, known as the Lower Confidence Bound (LCB) approach (Rashidinejad et al., 2021; Li et al., 2022; Shi et al., 2022; Yan et al., 2022). In this paper, we develop a direct approach to analyzing such uncertainty in the transition kernel by constructing a statistically plausible set of transition kernels, i.e., uncertainty set, and optimizing the worst-case performance over this uncertainty set. This principle is referred to as "distributionally robust optimization (DRO)" in the literature (Nilim & El Ghaoui, 2004; Iyengar, 2005). This DRO-based approach directly tackles the uncertainty in the transition kernel. We show that our approach asymptotically achieves the minimax optimal sample complexity (Rashidinejad et al., 2021). We summarize our major contributions as follows.

## 1.1 Main Contributions

In this work, we focus on the most general partial coverage setting (see Section 2.3 for the definition). We develop a DRO-based framework that efficiently solves the offline RL problem. More importantly, we design a value-function-based uncertainty set and show that its sample complexity is minimax optimal.

**DRO-based Approach Solves Offline RL.** We construct a distribution-based uncertainty set centered at the empirical transition kernel to guarantee that with high probability, the true transition kernel lies within the uncertainty set. Then, optimizing the worst-case performance over the uncertainty set provides a lower bound on the performance under the true environment. Our uncertainty model enables easy solutions using the robust dynamic programming approach developed for robust MDP in (Iyengar, 2005; Nilim & El Ghaoui, 2004) within a polynomial computational complexity. We further show the sample complexity to achieve an $\epsilon$-optimal policy using our approach is $\mathcal{O}\left(\frac{S^2 C^{\pi^*}}{(1-\gamma)^4 \epsilon^2}\right)$ (up to a log factor), where $\gamma$ is the discount factor, and $C^{\pi^*}$ is the single-policy concentrability for any comparator policy $\pi^*$ (see Definition 1). This sample complexity matches with the best-known complexity of the distribution-based model-uncertainty method (Rashidinejad et al., 2021; Uehara & Sun, 2021), which demonstrates that our DRO framework can directly tackle the model uncertainty and effectively solve offline RL.

**Achieving Asymptotic Minimax Optimality via Design of value-function-based Uncertainty Set.** While the approach described above is effective in achieving an $\epsilon$-optimal policy with relatively low sample complexity, it tends to exhibit excessive conservatism as its complexity surpasses the minimax lower bound for offline RL algorithms (Rashidinejad et al., 2021) by a factor of $S(1-\gamma)^{-1}$. To close this gap, we discover that demanding the true transition kernel to be within the uncertainty set with high probability, i.e., the true environment and the worst-case one are close, can be overly pessimistic and unnecessary. What is of paramount importance is that the value function under the worst-case transition kernel within the uncertainty set (almost) lower bounds the one under the true transition kernel. Notably, this requirement is considerably less stringent than mandating that the actual kernel be encompassed by the uncertainty set. We then design a less conservative value-function-based uncertainty set, which has a smaller radius and thus is a subset of the distribution-based uncertainty set. We prove that to obtain an $\epsilon$-optimal policy, the order of the sample complexity is $\mathcal{O}\left(\frac{S C^{\pi^*}}{(1-\gamma)^3 \epsilon^2}\right)$. This complexity indicates the asymptotic minimax optimality of our approach by matching with the minimax lower bound in asymptotic order of the sample complexity for offline RL (Rashidinejad et al., 2021) and the best result from the LCB approach (Li et al., 2022).

## 1.2 Related Works

There has been a proliferation of works on offline RL. In this section, we mainly discuss works on model-based approaches. There are also model-free approaches, e.g., (Liu et al., 2020; Kumar et al., 2020; Agarwal et al., 2020b; Yan et al., 2022; Shi et al., 2022), which are not the focus here.

**Offline RL under global coverage.** Existing studies on offline RL often make assumptions on the coverage of the dataset. This can be measured by the distribution shift between the behavior policy and the occupancy measure induced by the target policy, which is referred to as the concentrability coefficient (Munos, 2007; Rashidinejad et al., 2021). Many previous works, e.g., (Scherrer, 2014; Chen & Jiang, 2019; Jiang, 2019; Wang et al., 2019; Liao et al., 2020; Liu et al., 2019; Zhang et al., 2020a; Munos & Szepesvari, 2008; Uehara et al., 2020; Duan et al., 2020; Xie & Jiang, 2020; Levine et al., 2020; Antos et al., 2007; Farahmand et al., 2010), assume that the density ratio between the above two distributions is finite for all state-action pairs and policies, which is known as global coverage condition. This assumption essentially requires the behavior policy to be able to visit all possible state-action pairs, which is often violated in practice (Gulcehre et al., 2020; Agarwal et al., 2020b; Fu et al., 2020).

**Offline RL under partial coverage.** Recent studies relax the above assumption of global coverage to partial coverage or single-policy concentrability. Partial coverage assumes that the density ratio between the distributions induced by a single target policy and the behavior policy is bounded for all state-action pairs. Therefore, this assumption does not require the behavior policy to be able to visit all possible state-action pairs, as long as it can visit those state-actions pairs that the target policy will visit. This partial coverage assumption is more feasible and applicable in real-world scenarios. In this paper, we focus on this practical partial coverage setting. Existing approaches under the partial coverage assumption can be divided into three categories as follows.

- *Regularized Policy Search.* The first approach regularizes the policy so that the learned policy is close to the behavior policy (Fujimoto et al., 2019b; Wu et al., 2019; Jaques et al., 2019; Peng et al., 2019; Siegel et al., 2020; Wang et al., 2020; Kumar et al., 2019; Fujimoto et al., 2019a; Ghasemipour et al., 2020; Nachum et al., 2019; Zhang et al., 2020b; 2023). Thus, the learned policy is similar to the behavior policy which generates the dataset, hence this approach works well when the dataset is collected from experts (Wu et al., 2019; Fu et al., 2020).

- *Reward Penalization or LCB Approaches.* One of the most widely used approaches is to penalize the reward in face of uncertainty due to the limited data, to obtain an pessimistic estimation that lower bounds the real value function, e.g., (Kidambi et al., 2020; Yu et al., 2020; 2021b; Buckman et al., 2020; Jin et al., 2021; Xie et al., 2021b; Yin & Wang, 2021; Liu et al., 2020; Cui & Du, 2022; Chen et al., 2021; Zhong et al., 2022). The model-based approaches, e.g., VI-LCB (Rashidinejad et al., 2021; Li et al., 2022) penalizes the reward with a term that is inversely proportional to the number of samples. The tightest sample complexity is obtained in (Li et al., 2022) by designing a value-function-based penalty term, which matches the minimax lower bound showed in (Rashidinejad et al., 2021).

- *DRO-based Approaches.* Another approach is to first construct a set of "statistically plausible" MDP models based on the empirical transition kernel estimated using the dataset, and then find the policy that optimizes the worst-case performance over this set (Zanette et al., 2021; Uehara & Sun, 2021; Rigter et al., 2022; Bhardwaj et al., 2023; Guo et al., 2022; Hong et al., 2023; Chang et al., 2021; Blanchet et al., 2023). However, finding such a policy under the models proposed in these works can be NP-hard, hence some heuristic approximations without theoretical optimality guarantee are used to deploy their approaches. Our work falls into this category, but the computational complexity is polynomial, and the sample complexity of our approach is minimax optimal. A recent work (Panaganti et al., 2023) also proposes a distribution-based DRO framework similar to ours, and their sample complexity results match ours in the first part but fail to obtain the minimax optimality as our improved results in the second part.

- *Offline RL with function approximation.* There is another substantial interest in combining offline reinforcement learning with function approximation (e.g., deep neural networks) in order to encode

inductive biases and enable generalization across large, potentially continuous state spaces, with recent progress on both model-free and model-based approaches (Ross & Bagnell, 2012; Laroche et al., 2019; Fujimoto et al., 2019b; Kumar et al., 2019; Agarwal et al., 2020b). Besides the aforementioned challenges, an additional issue regarding the representational conditions of the function class is introduced, which assert that the function approximator is flexible enough to represent value functions induced by certain policies. A huge body of recent works aims to understand the condition for the function classes, including Bellman completeness condition (Liu et al., 2020; Jin et al., 2021; Xie et al., 2021a; Yin et al., 2021; Rashidinejad et al., 2021), or Bellman realizability (Xie & Jiang, 2021; Chen & Jiang, 2022; Rashidinejad et al., 2022; Jiang & Huang, 2020). However, in this paper, we mainly consider the tabular setting, where no requirement on the function class is needed.

**Robust RL with distributional uncertainty.** In this paper, our algorithm is based on the framework of robust MDP (Iyengar, 2005; Nilim & El Ghaoui, 2004; Bagnell et al., 2001; Satia & Lave Jr, 1973; Wiesemann et al., 2013), which finds the policy with the best worst-case performance over an uncertainty set of transition dynamics. When the uncertainty set is fully known, the problem can be solved by robust dynamic programming. The sample complexity of model-based approaches without full knowledge of the uncertainty sets was studied in, e.g., (Yang et al., 2021; Xu et al., 2023; Panaganti & Kalathil, 2022; Shi et al., 2023; Panaganti & Kalathil, 2022), where a generative model is typically assumed. This model-based approach is further adapted to the robust offline setting in (Panaganti et al., 2022; Shi & Chi, 2022). Yet in these works, the challenge induced by the offline setting is addressed using the LCB approach, i.e., penalizing the reward functions. In contrast, we show that the DRO framework itself addresses the challenge of partial coverage in the offline setting.

## 2 Preliminaries

### 2.1 Markov Decision Process (MDP)

An MDP can be characterized by a tuple $(\mathcal{S}, \mathcal{A}, \mathsf{P}, r)$, where $\mathcal{S}$ and $\mathcal{A}$ are the state and action spaces, $\mathsf{P} = \{\mathsf{P}_s^a \in \Delta(\mathcal{S}), a \in \mathcal{A}, s \in \mathcal{S}\}$[1] is the transition kernel, $r : \mathcal{S} \times \mathcal{A} \to [0,1]$ is the deterministic reward function[2], and $\gamma \in [0,1)$ is the discount factor. Specifically, $\mathsf{P}_s^a = (p_{s,s'}^a)_{s' \in \mathcal{S}}$, where $p_{s,s'}^a$ denotes the probability that the environment transits to state $s'$ if taking action $a$ at state $s$. The reward of taking action $a$ at state $s$ is given by $r(s,a)$. A stationary policy $\pi$ is a mapping from $\mathcal{S}$ to a distribution over $\mathcal{A}$, which indicates the probabilities of the agent taking actions at each state. At each time $t$, an agent takes an action $a_t \sim \pi(s_t)$ at state $s_t$, the environment then transits to the next state $s_{t+1}$ with probability $p_{s_t, s_{t+1}}^{a_t}$, and the agent receives reward $r(s_t, a_t)$.

The value function of a policy $\pi$ starting from any initial state $s \in \mathcal{S}$ is defined as the expected accumulated discounted reward by following $\pi$: $V_{\mathsf{P}}^\pi(s) \triangleq \mathbb{E}_{\mathsf{P}} \left[ \sum_{t=0}^\infty \gamma^t r(S_t, A_t) | S_0 = s, \pi \right]$, where $\mathbb{E}_{\mathsf{P}}$ denotes the expectation when the state transits according to $\mathsf{P}$. Let $\rho$ denote the initial state distribution, and denote the value function under the initial distribution $\rho$ by $V_{\mathsf{P}}^\pi(\rho) \triangleq \mathbb{E}_{s \sim \rho}[V_{\mathsf{P}}^\pi(s)]$.

### 2.2 Robust Markov Decision Process

In the robust MDP, the transition kernel is not fixed and lies in some uncertainty set $\mathcal{P}$. Define the robust value function of a policy $\pi$ as the worst-case expected accumulated discounted reward over the uncertainty set: $V_{\mathcal{P}}^\pi(s) \triangleq \min_{\mathsf{P} \in \mathcal{P}} \mathbb{E}_{\mathsf{P}} \left[ \sum_{t=0}^\infty \gamma^t r(S_t, A_t) | S_0 = s, \pi \right]$. Similarly, the robust action-value function for a policy $\pi$ is defined as $Q_{\mathcal{P}}^\pi(s,a) = \min_{\mathsf{P} \in \mathcal{P}} \mathbb{E}_{\mathsf{P}} \left[ \sum_{t=0}^\infty \gamma^t r(S_t, A_t) | S_0 = s, A_0 = a, \pi \right]$. The goal of robust RL is to find the optimal robust policy that maximizes the worst-case accumulated discounted reward, i.e., $\pi_r = \arg\max_\pi V_{\mathcal{P}}^\pi(s), \forall s \in \mathcal{S}$. It is shown in (Iyengar, 2005; Nilim & El Ghaoui, 2004; Wiesemann et al., 2013) that the optimal robust value function is the unique solution to the optimal robust Bellman equation

---

[1] $\Delta(\mathcal{S})$ denotes the probability simplex defined on $\mathcal{S}$.
[2] To streamline our presentation and emphasize our novelty in solving offline RL through DRO formulation, we assume the reward function is deterministic. However, our DRO approach directly extends to the stochastic reward setting by constructing a similar uncertainty set for the reward estimation.

$V_{\mathcal{P}}^{\pi_r}(s) = \max_a \{r(s,a) + \gamma \sigma_{\mathcal{P}_s^a}(V_{\mathcal{P}}^{\pi_r})\}$, where $\sigma_{\mathcal{P}_s^a}(V) \triangleq \min_{p \in \mathcal{P}_s^a} p^\top V$ denotes the support function of $V$ on a set $\mathcal{P}_s^a$ and the corresponding robust Bellman operator is a $\gamma$-contraction.

### 2.3 Offline Reinforcement Learning

Under the offline setting, the agent cannot interact with the MDP and instead is given a pre-collected dataset $\mathcal{D}$ consisting of $N$ tuples $\{(s_i, a_i, s_i', r_i) : i = 1, ..., N\}$, where $r_i = r(s_i, a_i)$ is the deterministic reward, and $s_i' \sim \mathsf{P}_{s_i}^{a_i}$ follows the transition kernel $\mathsf{P}$ of the MDP. The $(s_i, a_i)$ pairs in $\mathcal{D}$ are generated i.i.d. according to an unknown data distribution $\mu$ over the state-action space. In this paper, we consider the setting where the reward functions $r$ is deterministic but unknown. We denote the number of samples transitions from $(s,a)$ in $\mathcal{D}$ by $N(s,a)$, i.e., $N(s,a) = \sum_{i=1}^N \mathbf{1}_{(s_i,a_i)=(s,a)}$ and $\mathbf{1}_{X=x}$ is the indicator function.

The goal of offline RL is to find a policy $\pi$ which optimizes the value function $V_{\mathsf{P}}^\pi$ based on the offline dataset $\mathcal{D}$. Let $d^\pi$ denote the discounted occupancy distribution associated with $\pi$: $d^\pi(s) = (1-\gamma) \sum_{t=0}^\infty \gamma^t \mathbb{P}(S_t = s | S_0 \sim \rho, \pi, \mathsf{P})$. In this paper, we focus on the partial coverage setting and adopt the following definition from (Li et al., 2022) to measure the distribution shift between the dataset distribution and the occupancy measure induced by a deterministic single policy $\pi^*$:

**Definition 1.** *(Single-policy clipped concentrability) The single-policy clipped concentrability coefficient of a policy $\pi^*$ is defined as*

$$C^{\pi^*} \triangleq \max_{s,a} \frac{\min\{d^{\pi^*}(s,a), \frac{1}{S}\}}{\mu(s,a)}, \tag{1}$$

*where $S \triangleq |\mathcal{S}|$ denotes the number of states.*

We note another unclipped version of $C^{\pi^*}$ is also commonly used in the literature, e.g., (Rashidinejad et al., 2021; Uehara & Sun, 2021), defined as $\tilde{C}^{\pi^*} \triangleq \max_{s,a} \frac{d^{\pi^*}(s,a)}{\mu(s,a)}$. It is straightforward to verify that $C^{\pi^*} \leq \tilde{C}^{\pi^*}$, and all of our results remain valid if $C^{\pi^*}$ is replaced by $\tilde{C}^{\pi^*}$. A more detailed discussion can be found in (Li et al., 2022; Shi & Chi, 2022).

In this paper, we always assume that $C^{\pi^*} < \infty$ being a finite number, under which we aim to find a policy $\pi$ that minimizes the sub-optimality gap compared to a comparator policy $\pi^*$ under some initial state distribution $\rho$: $V_{\mathsf{P}}^{\pi^*}(\rho) - V_{\mathsf{P}}^\pi(\rho)$. It is worth noting that in some references, the comparator policy is set to be the optimal policy, but our results hold for arbitrary deterministic policies.

## 3 Offline RL via Distributionally Robust Optimization

Model-based methods usually commence by estimating the transition kernel employing its maximum likelihood estimate. Nevertheless, uncertainties can arise in these estimations due to the inherent challenges associated with distribution shifts and limited data in the offline setting. For instance, the dataset may not encompass every state-action pair, and the sample size may be insufficient to estimate the transition kernel accurately. In this paper, we directly quantify the uncertainty in the empirical estimation of the transition kernel and construct a set of "statistically possible" transition kernels, referred to as uncertainty sets, to encompass the actual environment. We then employ the DRO approach to optimize the worst-case performance over the uncertainty set. This formulation transforms the problem into a robust Markov Decision Process (MDP), as discussed in Section 2.2.

In Section 3.1, we first introduce a direct distribution-based approach to construct the uncertainty set such that the true transition kernel is in the uncertainty set with high probability. We then present the robust value iteration algorithm to solve the DRO problem. We further theoretically characterize the bound on the sub-optimality gap and show that the sample complexity to achieve an $\epsilon$-optimality gap is $\mathcal{O}((1-\gamma)^{-4}\epsilon^{-2}S^2 C^{\pi^*})$. This gap matches with the best-known sample complexity of the DRO-based approach in Uehara & Sun (2021), but with a polynomial complexity. This result shows the effectiveness of our DRO-based approach in solving the offline RL problem.

We then design a less conservative uncertainty set aiming to achieve the minimax optimal sample complexity in Section 3.2. We theoretically establish that our approach attains an enhanced and asymptotically minimax optimal sample complexity of $\mathcal{O}\left((1-\gamma)^{-3}\epsilon^{-2}SC^{\pi^*}\right)$. Notably, this sample complexity matches with the minimax lower bound (Rashidinejad et al., 2021) and stands on par with the best results achieved using the LCB approach (Li et al., 2022) in asymptotic order.

Our approach starts with learning an empirical model of the transition kernel and reward from the dataset as follows. These empirical transition kernels will be used as the centroid of the uncertainty set. For any $(s, a)$, if $N(s, a) > 0$, set

$$\hat{\mathsf{P}}^a_{s,s'} = \frac{\sum_{i \leq N} \mathbf{1}_{(s_i, a_i, s'_i)=(s,a,s')}}{N(s, a)}, \hat{r}(s, a) = r_i(s, a); \tag{2}$$

And if $N(s, a) = 0$, set

$$\hat{\mathsf{P}}^a_{s,s'} = \mathbf{1}_{s'=s}, \hat{r}(s, a) = 0. \tag{3}$$

The MDP $\hat{\mathsf{M}} = (\mathcal{S}, \mathcal{A}, \hat{\mathsf{P}}, \hat{r})$ with the empirical transition kernel $\hat{\mathsf{P}}$ and empirical reward $\hat{r}$ is referred to as the empirical MDP. For unseen state-action pairs $(s, a)$ in the offline dataset, we take a conservative approach and let the estimated $\hat{r}(s, a) = 0$, and set $s$ as an absorbing state if taking action $a$. Then the action value function at $(s, a)$ for the empirical MDP shall be zero, which discourages the choice of action $a$ at state $s$.

For each state-action pair $(s, a)$, we construct an uncertainty set centered at the empirical transition kernel $\hat{\mathsf{P}}^a_s$, with a radius inversely proportional to the number of samples in the dataset. Specifically, set the uncertainty set as $\hat{\mathcal{P}} = \bigotimes_{s,a} \hat{\mathcal{P}}^a_s$[3] and

$$\hat{\mathcal{P}}^a_s = \left\{q \in \Delta(\mathcal{S}) : D(q, \hat{\mathsf{P}}^a_s) \leq R^a_s\right\}, \tag{4}$$

where $D(\cdot, \cdot)$ is some function that measures the difference between two probability distributions, e.g., total variation, Chi-square divergence, $R^a_s$ is the radius ensuring that the uncertainty set adapts to the dataset size and the degree of confidence, which will be determined later. We then construct the robust MDP as $\hat{\mathsf{M}} = (\mathcal{S}, \mathcal{A}, \hat{\mathcal{P}}, \hat{r})$.

As we shall show later, the optimal robust policy

$$\pi_r = \arg\max_{\pi} \min_{\mathsf{P} \in \hat{\mathcal{P}}} V^{\pi}_{\mathsf{P}}(s), \forall s \in \mathcal{S}. \tag{5}$$

w.r.t. $\hat{\mathsf{M}}$ performs well in the real environment and reaches a small sub-optimality gap. In our construction in eq. (4), the uncertainty set is $(s, a)$-rectangular, i.e., for different state-action pairs, the corresponding uncertainty sets are independent. With this rectangular structure, the optimal robust policy can be found by utilizing the robust value iteration algorithm or robust dynamic programming (Algorithm 1), and the corresponding robust value iteration at each step can be solved in polynomial time (Wiesemann et al., 2013). In contrast, the uncertainty set constructed in (Uehara & Sun, 2021; Bhardwaj et al., 2023), defined as $\mathcal{T} = \{\mathsf{Q} : \mathbb{E}_{\mathcal{D}}[\|\hat{\mathsf{P}}^a_s - \mathsf{Q}^a_s\|^2] \leq \zeta\}$, does not enjoy such a rectangularity. Solving a robust MDP with such an uncertainty set can be, however, NP-hard (Wiesemann et al., 2013). The approach developed in (Rigter et al., 2022) to solve it is based on the heuristic approach of adversarial training, and therefore is lack of theoretical guarantee.

The algorithm converges to the optimal robust policy linearly since the robust Bellman operator is a $\gamma$-contraction (Iyengar, 2005). The computational complexity of the support function $\sigma_{\hat{\mathcal{P}}^a_s}(V)$ in Lines 3 and 7 w.r.t. the uncertainty sets we constructed matches the ones of the LCB approaches (Rashidinejad et al., 2021; Li et al., 2022).

In the following two sections, we specify the constructions of the uncertainty sets.

---

[3]Here, $\bigotimes_{s,a}$ denotes the Cartesian product of all state-action pairs, i.e., the uncertainty set $\hat{\mathcal{P}}$ is independently defined for each state-action pairs. This structure is also known as the $(s, a)$-rectangular uncertainty set.

---

**Algorithm 1** Robust Value Iteration (Nilim & El Ghaoui, 2004; Iyengar, 2005)

---

**INPUT**: $\hat{r}, \hat{\mathcal{P}}, V, \mathcal{D}$

  1: **while** TRUE **do**
  2:   **for** $s \in \mathcal{S}$ **do**
  3:     $V(s) \leftarrow \max_a \{\hat{r}(s, a) + \gamma \sigma_{\hat{\mathcal{P}}_s^a}(V)\}$
  4:   **end for**
  5: **end while**
  6: **for** $s \in \mathcal{S}$ **do**
  7:   $\pi_r(s) \in \arg\max_{a \in \mathcal{A}}\{\hat{r}(s, a) + \gamma \sigma_{\hat{\mathcal{P}}_s^a}(V)\}\}$
  8: **end for**

**Output**: $\pi_r$

---

### 3.1 Distribution-Based Radius

In this section, we first consider a straightforward, distribution-based construction of the uncertainty set. Namely, we aim to construct an uncertainty set such that the true transition kernel falls into it, such that the DRO framework can provide a lower bound or conservative estimation of the true performance.

We employ the total variation to construct this uncertainty set. Specifically, our uncertainty set is constructed as follows:

$$\hat{\mathcal{P}}_s^a = \left\{ q \in \Delta(\mathcal{S}) : \frac{1}{2}\|q - \hat{\mathsf{P}}_s^a\|_1 \leq R_s^a \triangleq \left\{ 1, \sqrt{\frac{S \log \frac{SA}{\delta}}{8N(s, a)}} \right\} \right\}. \tag{6}$$

With our design, fewer samples result in a larger uncertainty set and imply that we should be more conservative in estimating the transition dynamics at this state-action pair. Other distance function of $D$ can also be used, contingent upon the concentration inequality being applied.

In Algorithm 1, $\sigma_{\hat{\mathcal{P}}_s^a}(V) = \min_{q \in \hat{\mathcal{P}}_s^a}\{q^\top V\}$ can be equivalently solved by solving its dual form (Iyengar, 2005), which is a convex optimization problem: $\max_{0 \leq \mu \leq V}\{\hat{\mathsf{P}}_s^a(V - \mu) - R_s^a \mathrm{Span}(V - \mu)\}$, and $\mathrm{Span}(X) = \max_i X(i) - \min_i X(i)$ is the span semi-norm of vector $X$. The computational complexity associated with solving it is $\mathcal{O}(S \log(S))$. Notably, this polynomial computational complexity is on par with the complexity of the VI-LCB approach (Li et al., 2022).

We then show that with this distribution-based radius, the true transition kernel falls into the uncertainty set with high probability.

**Lemma 1.** *With probability at least $1 - \delta$, it holds that for any $s, a$, $\mathsf{P}_s^a \in \hat{\mathcal{P}}_s^a$, i.e., $\|\mathsf{P}_s^a - \hat{\mathsf{P}}_s^a\| \leq 2R_s^a$.*[4]

This result implies that the real environment $\mathsf{P}$ falls into the uncertainty set $\hat{\mathcal{P}}$ with high probability, and hence finding the optimal robust policy of $\hat{\mathcal{M}}$ provides a worst-case performance guarantee. We further present our result of the sub-optimality gap in the following theorem.

**Theorem 1.** *Consider an arbitrary deterministic comparator policy $\pi^*$. With probability at least $1 - 2\delta$, the output policy $\pi_r$ of Algorithm 1 satisfies*

$$V_{\mathsf{P}}^{\pi^*}(\rho) - V_{\mathsf{P}}^{\pi_r}(\rho) \leq \frac{16SC^{\pi^*}\log\frac{NS}{\delta}}{(1 - \gamma)^2 N} + \sqrt{\frac{96S^2 C^{\pi^*}\log\frac{SA}{\delta}}{(1 - \gamma)^4 N}}. \tag{7}$$

To achieve an $\epsilon$-optimality gap, a dataset of size $N = \mathcal{O}\left((1 - \gamma)^{-4}\epsilon^{-2}S^2 C^{\pi^*}\right)$ is required. This sample complexity matches with the sample complexity for LCB methods (Rashidinejad et al., 2021) and model uncertainty type approach (Panaganti et al., 2023; Uehara & Sun, 2021). It suggests that our DRO-based approach can effectively address the offline RL problem.

---

[4] In this paper, unless stated otherwise, we denote the $l_1$-norm by $\|\cdot\|$.

However, there is still a gap between this sample complexity and the minimax lower bound in (Rashidinejad et al., 2021) and the best-known sample complexity of LCB-based method (Li et al., 2022), which is $\mathcal{O}\left((1-\gamma)^{-3}\epsilon^{-2}SC^{\pi^*}\right)$. We will address this problem via a value function-based uncertainty set design in the next subsection.

**Remark 1.** *The choice of total variation for constructing the uncertainty set and obtain the results are not essential, and alternative distance functions or divergence, e.g., including Chi-square divergence, KL-divergence and Wasserstein distance can also be used with their corresponding concentration inequalities (Canonne, 2020; Bhandari & Russo, 2021; Arora et al., 2023). Our results and methods can be further generalized to large-scale problems when a low-dimensional latent representation is presented, e.g., linear MDPs or low-rank MDPs (Panaganti et al., 2023).*

### 3.2 Value-Function-Based Radius

As discussed above, using a distribution-based radius is able to achieve an $\epsilon$-optimal policy, however, with an unnecessarily large sample complexity. Compared with the minimax lower bound and the tightest result obtained in (Li et al., 2022), there exists a gap of order $\mathcal{O}(S(1-\gamma)^{-1})$. This gap is mainly because the distribution-based radius is overly conservative such that the true transition kernel $\mathsf{P}$ falls into $\hat{\mathcal{P}}$ with high probability (Lemma 1). Therefore, it holds that $V_{\hat{\mathcal{P}}}^{\pi_r} \leq V_{\mathsf{P}}^{\pi_r}$ and the sub-optimality gap can be bounded as

$$V_{\mathsf{P}}^{\pi^*} - V_{\mathsf{P}}^{\pi_r} = \underbrace{V_{\mathsf{P}}^{\pi^*} - V_{\hat{\mathcal{P}}}^{\pi_r}}_{\Delta_1} + \underbrace{V_{\hat{\mathcal{P}}}^{\pi_r} - V_{\mathsf{P}}^{\pi_r}}_{\Delta_2 \leq 0} \leq \Delta_1. \tag{8}$$

Since the radius is large, although $\Delta_2$ is bounded by 0, $V_{\hat{\mathcal{P}}}$ is more conservative and hence the first term $\Delta_1$ becomes larger, resulting in a loose bound with an order of $\mathcal{O}\left(\frac{1}{\sqrt{N(1-\gamma)^4}}\right)$.

This result can be improved from two aspects. Firstly, we note that the uncertainty set under the distribution-based construction is too large to include $\mathsf{P}$ with high probability. Although this construction implies $\Delta_2 \leq 0$, as the price of it, the large radius implies a loose bound on $\Delta_1$. Another observation is that both two terms are in fact the differences between the *expectations* under two different distributions. Instead of merely using the distance of the two distributions to bound them, we can utilize other tighter concentration inequalities like Bernstein's inequality, to obtain an involved but tighter bound.

Toward this goal, we construct a smaller and less conservative uncertainty set such that: (1). It implies a tighter bound on $\Delta_1$ combining with Bernstein's inequality; And (2). Although non-zero, the term $\Delta_2$ can also be bounded by some tight error bounds.

To do so, we further note that $\Delta_2 = \underbrace{V_{\hat{\mathcal{P}}}^{\pi_r} - V_{\hat{\mathsf{P}}}^{\pi_r}}_{(a)} + \underbrace{V_{\hat{\mathsf{P}}}^{\pi_r} - V_{\mathsf{P}}^{\pi_r}}_{(b)}$. Term $(b)$ can be viewed as an estimation error which is from the inaccurate estimation from the dataset, which is independent from the designing of the uncertain set (ignoring the dependence of $\pi_r$); And Term $(a)$ is the difference between robust value function and the value function under the centroid transition kernel $\hat{\mathsf{P}}$, which is always negative and can be bounded using the dual-form solutions for specific uncertainty sets (Iyengar, 2005). We hence choose a radius such that: (1). the negative bound on $(a)$ cancels with the higher-order terms in the bound on $(b)$ and further implies a tighter bound on $\Delta_2$; And (2). the bound on $\Delta_1$ is also tight by utilizing Bernstein's inequality.

Since our design is based on the value functions instead of distributions, we refer to it as the value-function-based construction. Our construction is as follows.

$$\hat{\mathcal{P}}_s^a = \left\{ q \in \Delta(\mathcal{S}) : \chi^2(q||\hat{\mathsf{P}}_s^a) \leq R_s^a \triangleq \frac{48 \log \frac{4SAN}{(1-\gamma)\delta}}{N(s,a)} \right\}. \tag{9}$$

Note that for state-action pairs $(s,a)$ with $N(s,a) = 0$, the uncertainty set reduces to the whole probability simplex $\Delta(\mathcal{S})$.

In our construction, instead of the total variation, we construct the uncertainty set using the Chi-square divergence, i.e., $D(p,q) = \chi^2(p\|q) = \sum_s q(s)\left(1 - \frac{p(s)}{q(s)}\right)^2$. As we will discuss later, adopting it can result in a tighter bound than total variation.

**Remark 2.** *From Pinsker's inequality and the fact that $D_{KL}(p\|q) \leq \chi^2(p\|q)$ (Nishiyama & Sason, 2020), it holds that $\|p-q\| \leq \sqrt{2\chi^2(p\|q)}$. Hence the value-function-based uncertainty set is a subset of the distribution-based uncertainty set in Section 3.1, verifying our value-function-based construction is less conservative.*

Similarly, we find the optimal robust policy w.r.t. the corresponding robust MDP $\hat{\mathcal{M}} = (\mathcal{S}, \mathcal{A}, \hat{\mathcal{P}}, \hat{r})$ using the robust value iteration with a slight modification, which is presented in Algorithm 2. Specifically, the output

---

**Algorithm 2** Robust Value Iteration

---

**INPUT**: $\hat{r}, \hat{\mathcal{P}}, V, \mathcal{D}$

1: **while** TRUE **do**
2:    **for** $s \in \mathcal{S}$ **do**
3:       $N(s) \leftarrow \sum_{i=1}^{N} \mathbf{1}_{(s_i=s)}$
4:       $V(s) \leftarrow \max_a \{\hat{r}(s,a) + \gamma\sigma_{\hat{\mathcal{P}}_s^a}(V)\}$
5:    **end for**
6: **end while**
7: **for** $s \in \mathcal{S}$ **do**
8:    **if** $N(s) > 0$ **then**
9:       $\pi_r(s) \in \{\arg\max_{a \in \mathcal{A}}\{\hat{r}(s,a) + \gamma\sigma_{\hat{\mathcal{P}}_s^a}(V)\}\} \cap \{a : N(s,a) > 0\}$
10:   **else**
11:      $\pi_r(s) \in \arg\max_{a \in \mathcal{A}}\{\hat{r}(s,a) + \gamma\sigma_{\hat{\mathcal{P}}_s^a}(V)\}\}$
12:   **end**
13: **end for**

**Output**: $\pi_r$

---

policy $\pi_r$ in Algorithm 2 is set to be the greedy policy satisfying $N(s,a) > 0$ if $N(s) > 0$. The existence of such a policy is proved in Lemma 2 in the appendix. This is to guarantee that when there is a tie of taking greedy actions, we will take an action that has appeared in the pre-collected dataset $\mathcal{D}$, align with the pessimism principle.

The support function $\sigma_{\hat{\mathcal{P}}_s^a}(V)$ w.r.t. the Chi-square divergence uncertainty set can also be computed using its dual form (Iyengar, 2005): $\sigma_{\hat{\mathcal{P}}_s^a}(V) = \max_{\alpha \in [V_{\min}, V_{\max}]}\{\hat{\mathsf{P}}_s^a V_\alpha - \sqrt{R_s^a \mathbf{Var}_{\hat{\mathsf{P}}_s^a}(V_\alpha)}\}$, where $V_\alpha(s) = \min\{\alpha, V(s)\}$. The dual form is also a convex optimization problem and can be solved efficiently within a polynomial time $\mathcal{O}(S \log S)$ (Iyengar, 2005).

**Remark 3.** *Using the Chi-square divergence enables a smaller radius and yields a tighter bound on $\Delta_2 = (a) + (b)$. Namely, (b) can be bounded by a $N^{-0.5}$-order bound according to the Bernstein's inequality (see Lemma 5 in the Appendix). Simultaneously, our goal is to obtain a bound with the same order on (a), which effectively offsets the bound on (b), and yields a tighter bound on $\Delta_2$. The robust value function w.r.t. the total variation uncertainty set, however, depends on $R_s^a$ linearly (see the dual form we discussed above); On the other hand, the solution to the Chi-square divergence uncertainty set incorporates a term of $\sqrt{R_s^a}$ which enables us to set a lower-order radius (i.e., set $R_s^a = (N(s,a))^{-1}$) to offset the $N^{-0.5}$-order bound on (b). Thus we choose Chi-square in our construction.*

We then characterize the optimality gap obtained from Algorithm 2 in the following theorem.

**Theorem 2.** *If $N \geq \mathcal{O}\left(\frac{1}{(1-\gamma)SC^{\pi^*}\mu_{\min}^2}\right)$, then the output policy $\pi_r$ of Algorithm 2 satisfies*

$$V_{\mathsf{P}}^{\pi^*}(\rho) - V_{\mathsf{P}}^{\pi_r}(\rho) \leq \sqrt{\frac{KSC^{\pi^*}\log\frac{4SAN}{(1-\gamma)\delta}}{(1-\gamma)^3 N}}, \tag{10}$$

*with probability at least $1 - 4\delta$, where $\mu_{\min} = \min\{\mu(s, a) : \mu(s, a) > 0\}$ denotes the minimal non-zero entry of $\mu$, and $K$ is some universal constant that is independent from $S, \gamma, C^{\pi^*}$ and $N$.*

Theorem 2 implies that our DRO approach can achieve an $\epsilon$-optimality gap, as long as the size of the dataset exceeds the order of

$$\mathcal{O}\left(\underbrace{\frac{SC^{\pi^*}}{(1-\gamma)^3\epsilon^2}}_{\epsilon\text{-dependent}} + \underbrace{\frac{1}{(1-\gamma)SC^{\pi^*}\mu_{\min}^2}}_{\text{burn-in cost}}\right). \tag{11}$$

The burn-in cost term indicates that the asymptotic bound of the sample complexity becomes relevant after the dataset size surpasses the burn-in cost. It represents the minimal requirement for the amount of data. In fact, if the dataset is too small, we should not expect to learn a well-performed policy from it. In our case, if the dataset is generated under a generative model Panaganti et al. (2022); Yang et al. (2021); Shi et al. (2023) or uniform distribution, the burn-in cost term is in order of $\frac{SA^2}{1-\gamma}$. Burn-in cost also widely exists in the sample complexity studies of RL, e.g., $H^8 SC^{\pi^*}$ in (Xie et al., 2021b), $\frac{S^3A^2}{(1-\gamma)^4}$ in (He et al., 2021), and $\frac{SC^{\pi^*}}{(1-\gamma)^5}$ in (Yan et al., 2022), $\frac{H}{\mu_{\min}p_{\min}}$ in (Shi & Chi, 2022). Note that the burn-in cost term is independent of the accuracy level $\epsilon$, which implies the sample complexity is less than $\mathcal{O}\left(\frac{SC^{\pi^*}}{(1-\gamma)^3\epsilon^2}\right)$, as long as $\epsilon$ is small. This result matches the optimal asymptotic order of the complexity according to the minimax lower bound in (Rashidinejad et al., 2021), and also matches the tightest bound obtained using the LCB approach (Li et al., 2022) asymptotically. This suggests that our DRO approach can effectively address offline RL while imposing minimal demands on the dataset, thus optimizing the sample complexity associated with offline RL.

### 3.3 Further Comparison with LCB Approaches

We further discuss the major differences in our approach and the LCB approaches (Li et al., 2022; Rashidinejad et al., 2021) in this section.

Firstly, the motivations of the two approaches are different. The LCB approach can be viewed as **'value function-based'**, which aims to obtain a pessimistic estimation of the value function by subtracting a penalty term from the reward, e.g., eq (83) in (Li et al., 2022). It implies that the resulting estimation lower bounds the true value functions. Our DRO approach aims to construct an uncertainty set that contains the statistically plausible transition dynamics, and optimize the worst-case performance among this uncertainty set, which can be viewed as **'transition model-based'**, meaning that we directly tackle the uncertainty from the model estimation, without using the value function as an intermediate step. The robust value function we obtained, may not be a conservative estimation of the true value functions, but it utilizes the principle of pessimism through the DRO formulation.

Our proof techniques are also different from the ones in LCB. To clarify the difference, we first rewrite our update rule using the LCB fashion. The update in robust value iteration Algorithm 1 can be written as

$$V(s) \leftarrow \max_a\{r(s, a) + \gamma\sigma_{\hat{\mathcal{P}}_s^a}(V)\} = \max_a\{r(s, a) + \gamma\hat{\mathsf{P}}_s^a V - b(s, a, V)\}, \tag{12}$$

where $b(s, a, V) \triangleq \gamma\hat{\mathsf{P}}_s^a V - \gamma\sigma_{\hat{\mathcal{P}}_s^a}(V)$, and hence, our algorithm bears formal resemblance to an LCB approach.

However, this formal similarity does not imply that our approach and results can be derived from those of LCB methods. Specifically, in LCB approaches, a crucial step involves the meticulous design of the penalty term $b(s, a, V)$ to satisfy the condition $\gamma|\hat{\mathsf{P}}_s^a V - \mathsf{P}_s^a V| \leq b(s, a, V)$, ensuring that the obtained estimation $V$ remains pessimistic compared to the true value function. This often leads to the development of a complex penalty term, which may incorporate variance (e.g., see equation (28) in (Li et al., 2022)).

In contrast, within the DRO setting, designing an uncertainty set to uphold the aforementioned inequality either yields a loose bound on sub-optimality or results in a highly intricate uncertainty set. On one hand, enlarging the uncertainty set sufficiently to accommodate $\mathsf{P} \in \hat{\mathcal{P}}$ (to ensure $\mathsf{P}_s^a V \geq \min_{q \in \hat{\mathcal{P}}_s^a} qV = \sigma_{\hat{\mathcal{P}}_s^a}(V)$ and the inequality above) falls into the distribution-based construction, leading to sub-optimal sample complexity.

On the other hand, adopting a similar approach involving Bernstein inequality introduces dependence on $V$ into the radius, resulting in a time-varying and intricate uncertainty set akin to the one in the LCB method. Our method circumvents these issues by not mandating the above inequality for leveraging the pessimism principle. Instead, we harness the inherent pessimism of the DRO setting and devise a straightforward yet effective uncertainty set. Although the above inequality may not hold, the robust value function we derive remains a pessimistic estimation of the true value function, demonstrating its effectiveness and efficiency.

This underscores the fundamental disparity in motivation and design between our method and LCB approaches, highlighting the novelty of our approach.

We present a comparative analysis of our results alongside those of the most closely related works in Table 1. As evidenced by the comparison, our approach stands out as the first model-uncertainty-based method to achieve the minimax optimal sample complexity in offline RL. Notably, we observe variations in the assumptions made across these works, particularly concerning the adoption of the clipped version of the single-policy concentrability. We meticulously specify these variations in our comparison. However, as previously discussed, the single-policy clipped concentrability is inherently smaller than its unclipped counterpart. Consequently, our assumption is comparatively weaker, enabling our results to directly follow if we adopt the unclipped assumption.

| | Approach Type | Assumption | Asymptotic Sample Complexity | Computational Complexity |
|---|---|---|---|---|
| Our Approach | DRO | Single-policy, clipped | $\mathcal{O}\left(\frac{SC^{\pi^*}}{\epsilon^2(1-\gamma)^3}\right)$ | Polynomial |
| (Rashidinejad et al., 2021) | LCB | Single-policy | $\mathcal{O}\left(\frac{SC^{\pi^*}}{\epsilon^2(1-\gamma)^5}\right)$ | Polynomial |
| (Uehara & Sun, 2021) | DRO | Single-policy | $\mathcal{O}\left(\frac{S^2C^{\pi^*}}{\epsilon^2(1-\gamma)^4}\right)$ | NP-Hard |
| (Panaganti et al., 2023) | DRO | Single-policy | $\mathcal{O}\left(\frac{S^2C^{\pi^*}}{\epsilon^2(1-\gamma)^4}\right)$ | Polynomial |
| (Li et al., 2022) | LCB | Single-policy, clipped | $\mathcal{O}\left(\frac{SC^{\pi^*}}{\epsilon^2(1-\gamma)^3}\right)$ | Polynomial |
| (Rashidinejad et al., 2021) | Minimax Lower bound | Single-policy | $\mathcal{O}\left(\frac{SC^{\pi^*}}{\epsilon^2(1-\gamma)^3}\right)$ | - |

Table 1: Comparison with related works.

## 4 Experiments

We adapt our DRO framework under two problems, the Garnet problem $\mathcal{G}(30, 20)$ (Archibald et al., 1995), and the Frozen-Lake problem (Brockman et al., 2016) to numerically verify our results.

In the Garnet problem, $|\mathcal{S}| = 30$ and $|\mathcal{A}| = 20$. The transition kernel $\mathsf{P} = \{\mathsf{P}_s^a, s \in \mathcal{S}, a \in \mathcal{A}\}$ is randomly generated following a normal distribution: $\mathsf{P}_s^a \sim \mathcal{N}(\omega_s^a, \sigma_s^a)$ and then normalized, and the reward function $r(s, a) \sim \mathcal{N}(\nu_s^a, \psi_s^a)$, where $\omega_s^a, \sigma_s^a, \nu_s^a, \psi_s^a \sim \mathbf{Uniform}[0, 100]$.

In the Frozen-Lake problem, an agent aim to cross a $4 \times 4$ frozen lake from Start to Goal without falling into any Holes by walking over the frozen lake.

In both problems, we deploy our approach under both global coverage and partial coverage conditions. Specifically, under the global coverage setting, the dataset is generated by the uniform policy $\pi(a|s) = \frac{1}{|\mathcal{A}|}$; And under the partial coverage condition, the dataset is generated according to $\mu(s, a) = \frac{\mathbf{1}_{a=\pi^*(s)}}{2} + \frac{\mathbf{1}_{a=\eta}}{2}$, where $\eta$ is an action randomly chosen from the action space $\mathcal{A}$.

At each time step, we generate 40 new samples and add them to the offline dataset and deploy our DRO approach on it. We also deploy the LCB approach (Li et al., 2022) and non-robust model-based dynamic programming as the baselines. We run the algorithms independently 10 times and plot the average value of the sub-optimality gaps over all 10 trajectories. We also plot the 95th and 5th percentiles of the 10 curves as the upper and lower envelopes of the curves. The results are presented in Figure 1. It can be seen from the results that our DRO approach finds the optimal policy with relatively less data; The LCB approach has a similar convergence rate to the optimal policy, which verifies our theoretical results; The non-robust DP converges much slower, and can even converge to a sub-optimal policy. The results hence demonstrate the effectiveness and efficiency of our DRO approach.

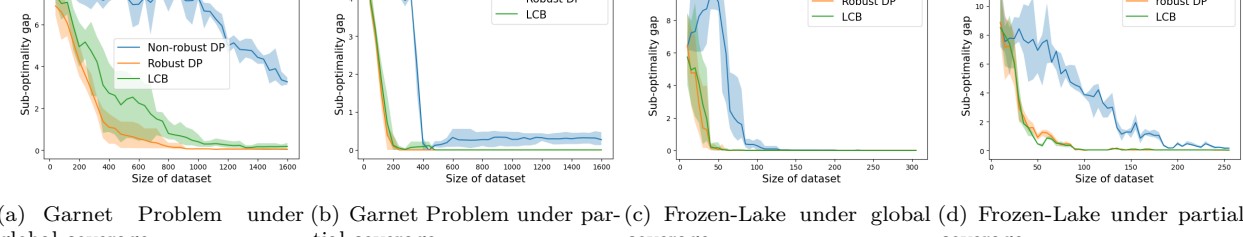

(a) Garnet Problem under global coverage

(b) Garnet Problem under partial coverage

(c) Frozen-Lake under global coverage

(d) Frozen-Lake under partial coverage

Figure 1: Sub-optimality gaps of Robust DP, LCB approach, and Non-robust DP.

## 5 Conclusion

In this paper, we revisit the problem of offline reinforcement learning from a novel angle of distributional robustness. We develop a DRO-based approach to solve offline reinforcement learning. Our approach directly incorporates conservatism in estimating the transition dynamics instead of penalizing the reward of less-visited state-action pairs. Our algorithms are based on the robust dynamic programming approach, which is computationally efficient. We focus on the challenging partial coverage setting, and develop two uncertainty sets: the distribution-based and the less conservative value-function-based. For the distribution-based uncertainty set, we theoretically characterize its sample complexity, and show it matches with the best one of the LCB-based approaches using the distribution-based bonus term. For the value-function-based uncertainty set, we show its sample complexity is asymptotically minimax optimal, with a constant-level burn-in cost. Our results provide a DRO-based framework, an alternative approach to efficiently and effectively solve the problem of offline reinforcement learning.

## 6 Acknowledgement

This work is supported by National Science Foundation under Grants CCF-2007783, CCF-2106560 and ECCS-2337375 (CAREER).

This material is based upon work supported under the AI Research Institutes program by National Science Foundation and the Institute of Education Sciences, U.S. Department of Education through Award # 2229873 - National AI Institute for Exceptional Education. Any opinions, findings and conclusions or recommendations expressed in this material are those of the author(s) and do not necessarily reflect the views of the National Science Foundation, the Institute of Education Sciences, or the U.S. Department of Education.

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

## A    Notations

We first introduce some notations that are used in our proofs. We denote the numbers of states and actions by $S, A$, i.e., $|\mathcal{S}| = S$, $|\mathcal{A}| = A$. For a transition kernel $\mathsf{P}$ and a policy $\pi$, $\mathsf{P}^\pi$ denotes the transition matrix induced by them, i.e., $\mathsf{P}^\pi(s) = \sum_a \pi(a|s)\mathsf{P}_s^a \in \Delta(\mathcal{S})$.

For any vector $V \in \mathbb{R}^S$, $V \circ V \in \mathbb{R}^S$ denotes the entry-wise multiplication, i.e., $V \circ V(s) = V(s) * V(s)$. For a distribution $q \in \Delta(\mathcal{S})$, it is straightforward to verify that the variance of $V$ w.r.t. $q$ can be rewritten as $\mathbf{Var}_q(V) = q(V \circ V) - (qV)^2$.

## B    Proofs of Section 3.1

In this section, we present our proofs of the results under the distribution-based construction.

We first note that Lemma 1 can be directly obtained from the existing results of concentration inequality, e.g., Theorem 1 from (Canonne, 2020) or Theorem 2.2 from (Weissman et al., 2003). Hence the proof is omitted here.

**Theorem 3.** *(Restatement of Theorem 1) With probability at least $1 - 2\delta$, it holds that*

$$V_\mathsf{P}^{\pi^*}(\rho) - V_\mathsf{P}^{\pi_r}(\rho) \leq \frac{2}{(1-\gamma)^2} \frac{8SC^{\pi^*} \log \frac{NS}{\delta}}{N} + \frac{2}{(1-\gamma)^2} \sqrt{\frac{24S^2 C^{\pi^*} \log \frac{SA}{\delta}}{N}}, \tag{13}$$

*To obtain an $\epsilon$-optimal policy, a dataset of size*

$$N = \mathcal{O}\left(\frac{SC^{\pi^*}}{(1-\gamma)^4 \epsilon^2}\right) \tag{14}$$

*is required.*

*Proof.* In the following proof, we only focus on the case when

$$N > \frac{8SC^{\pi^*} \log \frac{NS}{\delta}}{1 - \gamma}; \tag{15}$$

Otherwise, eq. (7) follows directly from the trivial bound $V_\mathsf{P}^{\pi^*}(\rho) - V_\mathsf{P}^{\pi_r}(\rho) \leq \frac{1}{1-\gamma}$.

According to Lemma 1, with probability at least $1 - \delta$, $\mathsf{P} \in \hat{\mathcal{P}}$. Moreover, due to the fact $r(s, a) \geq \hat{r}(s, a)$, hence

$$V_\mathsf{P}^\pi(s) \geq V_{\hat{r},\mathsf{P}}^\pi(s) \geq V_{\hat{r},\hat{\mathcal{P}}}^\pi(s) = V_{\hat{\mathcal{P}}}^\pi(s) \tag{16}$$

for any $\pi$ and $s \in \mathcal{S}$, where $V_{\hat{r},\mathsf{P}}^\pi$ denotes the value function w.r.t. $\mathsf{P}$ and reward $\hat{r}$. Thus $V_\mathsf{P}^\pi \geq V_{\hat{\mathcal{P}}}^\pi$ for any policy $\pi$.

Therefore,

$$\begin{aligned}
& V_\mathsf{P}^{\pi^*}(s) - V_\mathsf{P}^{\pi_r}(s) \\
&= V_\mathsf{P}^{\pi^*}(s) - V_{\hat{\mathcal{P}}}^{\pi_r}(s) + V_{\hat{\mathcal{P}}}^{\pi_r}(s) - V_\mathsf{P}^{\pi_r}(s) \\
&\leq V_\mathsf{P}^{\pi^*}(s) - V_{\hat{\mathcal{P}}}^{\pi_r}(s) \\
&= r(s, \pi^*(s)) + \gamma \mathsf{P}_s^{\pi^*(s)} V_\mathsf{P}^{\pi^*} - V_{\hat{\mathcal{P}}}^{\pi_r}(s) \\
&\overset{(a)}{\leq} r(s, \pi^*(s)) - \hat{r}(s, \pi^*(s)) + \gamma \mathsf{P}_s^{\pi^*(s)} V_\mathsf{P}^{\pi^*} - \gamma \sigma_{\hat{\mathcal{P}}_s^{\pi^*(s)}}(V_{\hat{\mathcal{P}}}^{\pi_r}) \\
&= r(s, \pi^*(s)) - \hat{r}(s, \pi^*(s)) + \gamma \mathsf{P}_s^{\pi^*(s)} V_\mathsf{P}^{\pi^*} - \gamma \mathsf{P}_s^{\pi^*(s)} V_{\hat{\mathcal{P}}}^{\pi_r} + \gamma \mathsf{P}_s^{\pi^*(s)} V_{\hat{\mathcal{P}}}^{\pi_r} - \gamma \sigma_{\hat{\mathcal{P}}_s^{\pi^*(s)}}(V_{\hat{\mathcal{P}}}^{\pi_r}) \\
&= r(s, \pi^*(s)) - \hat{r}(s, \pi^*(s)) + \gamma \mathsf{P}_s^{\pi^*(s)}(V_\mathsf{P}^{\pi^*} - V_{\hat{\mathcal{P}}}^{\pi_r}) + \gamma(\mathsf{P}_s^{\pi^*(s)} V_{\hat{\mathcal{P}}}^{\pi_r} - \sigma_{\hat{\mathcal{P}}_s^{\pi^*(s)}}(V_{\hat{\mathcal{P}}}^{\pi_r}))
\end{aligned}$$

$$\triangleq \gamma \mathsf{P}_s^{\pi^*(s)}(V_{\mathsf{P}}^{\pi^*} - V_{\hat{\mathcal{P}}}^{\pi_r}) + b^*(V_{\hat{\mathcal{P}}}^{\pi_r}), \tag{17}$$

where $(a)$ is from $V_{\hat{\mathcal{P}}}^{\pi_r}(s) = \max_a Q_{\hat{\mathcal{P}}}^{\pi_r}(s,a) \geq Q_{\hat{\mathcal{P}}}^{\pi_r}(s,\pi^*(s)) = \hat{r}(s,\pi^*(s)) + \gamma\sigma_{\hat{\mathcal{P}}_s^{\pi^*(s)}}(V_{\hat{\mathcal{P}}}^{\pi_r})$, and $b^*(V)(s) \triangleq$
$r(s,\pi^*(s)) - \hat{r}(s,\pi^*(s)) + \gamma\mathsf{P}_s^{\pi^*(s)}V - \gamma\sigma_{\hat{\mathcal{P}}_s^{\pi^*(s)}}(V)$.

Recursively applying this inequality further implies

$$V_{\mathsf{P}}^{\pi^*}(\rho) - V_{\mathsf{P}}^{\pi_r}(\rho) \leq \frac{1}{1-\gamma}\left\langle d^{\pi^*}, b^*(V_{\hat{\mathcal{P}}}^{\pi_r})\right\rangle, \tag{18}$$

where $d^{\pi^*}(\cdot) = (1-\gamma)\sum_{t=0}^\infty \gamma^t \mathbb{P}(S_t = \cdot|S_0 \sim \rho, \pi^*, \mathsf{P})$ is the discounted visitation distribution induced by $\pi^*$ and $\mathsf{P}$.

To bound the term in eq. (17), we introduce the following notations.

$$\mathcal{S}_s \triangleq \left\{s : N\mu(s,\pi^*(s)) \leq 8\log\frac{NS}{\delta}\right\}, \tag{19}$$

$$\mathcal{S}_l \triangleq \left\{s : N\mu(s,\pi^*(s)) > 8\log\frac{NS}{\delta}\right\}. \tag{20}$$

The two sets divide the state space into two sub-spaces. $\mathcal{S}_s$ contains the states that have less probability in the behavior distribution $\mu$ on its corresponding optimal action, and are less-frequently covered by the dataset; And $\mathcal{S}_l$ are the ones that have larger probability, and are more frequently covered.

For $s \in \mathcal{S}_s$, from eq. (15), we have that

$$\min\left\{d^{\pi^*}(s), \frac{1}{S}\right\} \leq C^{\pi^*}\mu(s,\pi^*(s)) \leq \frac{8C^{\pi^*}\log\frac{NS}{\delta}}{N} < \frac{1}{S}, \tag{21}$$

which further implies that $d^{\pi^*}(s) \leq \frac{8C^{\pi^*}\log\frac{NS}{\delta}}{N}$. Hence

$$\sum_{s\in\mathcal{S}_s} d^{\pi^*}(s)b^*(V_{\hat{\mathcal{P}}}^{\pi_r})(s) \leq \frac{1}{1-\gamma}\frac{8SC^{\pi^*}\log\frac{NS}{\delta}}{N}, \tag{22}$$

which is due to $b^*(V)(s) \triangleq r(s,\pi^*(s)) - \hat{r}(s,\pi^*(s)) + \gamma\mathsf{P}_s^{\pi^*(s)}V - \gamma\sigma_{\hat{\mathcal{P}}_s^{\pi^*(s)}}(V) \leq 1 + \frac{\gamma}{1-\gamma} = \frac{1}{1-\gamma}$.

We then consider $s \in \mathcal{S}_l$. From the definition, $N\mu(s,\pi^*(s)) > 8\log\frac{NS}{\delta}$. According to Lemma 4, with probability $1-\delta$,

$$\max\{12N(s,\pi^*(s)), 8\log\frac{NS}{\delta}\} \geq N\mu(s,\pi^*(s)) > 8\log\frac{NS}{\delta}, \tag{23}$$

hence $\max\{12N(s,\pi^*(s)), 8\log\frac{NS}{\delta}\} = 12N(s,\pi^*(s))$ and $N(s,\pi^*(s)) \geq \frac{2}{3}\log\frac{NS}{\delta} > 0$. This hence implies that for any $s \in \mathcal{S}_l$, $N(s,\pi^*(s)) > 0$ and $\hat{r}(s,\pi^*(s)) = r(s,\pi^*(s))$. Thus

$$|b^*(V_{\hat{\mathcal{P}}}^{\pi_r})(s)| = \gamma|\mathsf{P}_s^{\pi^*(s)}V_{\hat{\mathcal{P}}}^{\pi_r} - \sigma_{\hat{\mathcal{P}}_s^{\pi^*(s)}}(V_{\hat{\mathcal{P}}}^{\pi_r})|$$

$$\leq \|\mathsf{P}_s^{\pi^*(s)} - \mathsf{Q}_s^{\pi^*(s)}\|_1\|V_{\hat{\mathcal{P}}}^{\pi_r}\|_\infty$$

$$\leq \frac{2}{1-\gamma}\min\left\{2, \sqrt{\frac{S\log\frac{SA}{\delta}}{2N(s,\pi^*(s))}}\right\}$$

$$\leq \frac{1}{1-\gamma}\sqrt{\frac{2S\log\frac{SA}{\delta}}{N(s,\pi^*(s))}}. \tag{24}$$

Moreover, from eq. (23),

$$\frac{1}{N(s,\pi^*(s))} \leq \frac{12}{N\mu(s,\pi^*(s))} \leq \frac{12C^{\pi^*}}{N\min\{d^{\pi^*}(s),\frac{1}{S}\}} \leq \frac{12C^{\pi^*}}{N}\left(\frac{1}{d^{\pi^*}(s)} + S\right). \tag{25}$$

Combining with eq. (24) further implies

$$|b^*(V_{\hat{\mathcal{P}}}^{\pi_r})(s)| \leq \frac{1}{1-\gamma}\sqrt{2S\log\frac{SA}{\delta}}\sqrt{\frac{12C^{\pi^*}}{N}\left(\frac{1}{d^{\pi^*}(s)}+S\right)}$$

$$\leq \frac{1}{1-\gamma}\sqrt{2S\log\frac{SA}{\delta}}\sqrt{\frac{12C^{\pi^*}}{N}\frac{1}{d^{\pi^*}(s)}}+\frac{1}{1-\gamma}\sqrt{2S\log\frac{SA}{\delta}}\sqrt{\frac{12C^{\pi^*}}{N}S}. \qquad (26)$$

Thus

$$\sum_{s\in\mathcal{S}_l}d^{\pi^*}(s)b^*(V_{\hat{\mathcal{P}}}^{\pi_r})(s) \leq \frac{1}{1-\gamma}\sqrt{\frac{24SC^{\pi^*}\log\frac{SA}{\delta}}{N}}\sum_s\sqrt{d^{\pi^*}(s)}+\frac{1}{1-\gamma}\sqrt{\frac{24S^2C^{\pi^*}\log\frac{SA}{\delta}}{N}}$$

$$\leq \frac{1}{1-\gamma}\sqrt{\frac{24S^2C^{\pi^*}\log\frac{2}{\delta}}{N}}+\frac{1}{1-\gamma}\sqrt{\frac{24S^2C^{\pi^*}\log\frac{SA}{\delta}}{N}}$$

$$= \frac{2}{1-\gamma}\sqrt{\frac{24S^2C^{\pi^*}\log\frac{SA}{\delta}}{N}}. \qquad (27)$$

Thus combining eq. (22) and eq. (27) implies

$$V_{\mathsf{P}}^{\pi^*}(\rho) - V_{\mathsf{P}}^{\pi_r}(\rho)$$

$$\leq \frac{1}{1-\gamma}\left\langle d^{\pi^*}, b^*(V_{\hat{\mathcal{P}}}^{\pi_r})\right\rangle$$

$$\leq \frac{2}{(1-\gamma)^2}\frac{8SC^{\pi^*}\log\frac{NS}{\delta}}{N}+\frac{1}{(1-\gamma)^2}\sqrt{\frac{24S^2C^{\pi^*}\log\frac{SA}{\delta}}{N}}, \qquad (28)$$

which completes the proof. □

## C  Proofs of Section 3.2

In this section, we provide a refined analysis of the sub-optimality gap using Bernstein's Inequality.

### C.1  Existence of the Target Policy

In this section, we first prove the claim we made after Alg 2, i.e., there exists an optimal robust policy $\pi$ to the constructed uncertainty set $\hat{\mathcal{P}}$, such that $N(s,\pi(s)) > 0, \forall s \in \mathcal{S}$.

**Lemma 2.** *Recall the set $\mathcal{S}^0 \triangleq \{s \in \mathcal{S} : N(s) = 0\}$. Then*

*(1). For any policy $\pi$ and $s \in \mathcal{S}^0$, $V_{\hat{\mathcal{P}}}^{\pi}(s) = 0$;*

*(2). There exists a deterministic robust optimal policy $\pi_r$, such that for any $s \notin \mathcal{S}^0$, $N(s,\pi_r(s)) > 0$.*

*Proof.* **Proof of (1).**

For any $s \in \mathcal{S}^0$, it holds that $N(s,a) = 0$ for any $a \in \mathcal{A}$. Hence $\hat{r}(s,a) = 0$ and $\hat{\mathcal{P}}_s^a = \Delta(\mathcal{S})$.

Then for any policy $\pi$ and $a \in \mathcal{A}$, it holds that

$$Q_{\hat{\mathcal{P}}}^{\pi}(s,a) = \hat{r}(s,a) + \gamma\sigma_{\hat{\mathcal{P}}_s^a}(V_{\hat{\mathcal{P}}}^{\pi}) = \gamma\sigma_{\Delta(\mathcal{S})}(V_{\hat{\mathcal{P}}}^{\pi}) \leq \gamma V_{\hat{\mathcal{P}}}^{\pi}(s). \qquad (29)$$

Thus

$$V_{\hat{\mathcal{P}}}^{\pi}(s) = \sum_a \pi(a|s)Q_{\hat{\mathcal{P}}}^{\pi}(s,a) \leq \gamma V_{\hat{\mathcal{P}}}^{\pi}(s), \qquad (30)$$

which implies $V_{\hat{\mathcal{P}}}^{\pi}(s) = 0$ together with $V_{\hat{\mathcal{P}}}^{\pi} \geq 0$.

**Proof of (2).**

We prove Claim (2) by contradiction. Assume that for any optimal policy $\pi_r$, there exists $s \notin \mathcal{S}^0$ such that $N(s, \pi_r(s)) = 0$. We then consider a fixed pair $(\pi_r, s)$.

$N(s, \pi_r(s)) = 0$ further implies $\hat{r}(s, \pi_r(s)) = 0$, $\hat{\mathcal{P}}_s^{\pi_r(s)} = \Delta(\mathcal{S})$, and

$$V_{\hat{\mathcal{P}}}^{\pi_r}(s) = \max_a Q_{\hat{\mathcal{P}}}^{\pi_r}(s, a) = Q_{\hat{\mathcal{P}}}^{\pi_r}(s, \pi_r(s)) = \hat{r}(s, \pi_r(s)) + \gamma \sigma_{\hat{\mathcal{P}}_s^{\pi_r(s)}}(V_{\hat{\mathcal{P}}}^{\pi_r}) \leq \gamma V_{\hat{\mathcal{P}}}^{\pi_r}(s), \tag{31}$$

where the last inequality is from $\hat{\mathcal{P}}_s^{\pi_r(s)} = \Delta(\mathcal{S})$, $\hat{r}(s, \pi_r(s)) = 0$, and $\sigma_{\hat{\mathcal{P}}_s^{\pi_r(s)}}(V_{\hat{\mathcal{P}}}^{\pi_r}) \leq \mathbf{1}_s V_{\hat{\mathcal{P}}}^{\pi_r} = V_{\hat{\mathcal{P}}}^{\pi_r}(s)$. This further implies that $V_{\hat{\mathcal{P}}}^{\pi_r}(s) = 0$ because $V_{\hat{\mathcal{P}}}^{\pi_r} \geq 0$.

On the other hand, since $s \notin \mathcal{S}^0$, there exists another action $b \neq \pi_r(s)$ such that $N(s, b) > 0$, and hence $\hat{r}(s, b) = r(s, b)$. There can only be two cases as follows.

(I). If $r(s, b) > 0$, then

$$Q_{\hat{\mathcal{P}}}^{\pi_r}(s, b) = \hat{r}(s, b) + \gamma \sigma_{\hat{\mathcal{P}}_s^b}(V_{\hat{\mathcal{P}}}^{\pi_r}) > 0 = Q_{\hat{\mathcal{P}}}^{\pi_r}(s, \pi_r(s)), \tag{32}$$

which is contradict to $V_{\hat{\mathcal{P}}}^{\pi_r}(s) = \max_a Q_{\hat{\mathcal{P}}}^{\pi_r}(s, a) = Q_{\hat{\mathcal{P}}}^{\pi_r}(s, \pi_r(s))$.

(II). If $r(s, b) = 0$, Lemma 3 then implies the modified policy $f_b^s(\pi_r)$ is also optimal, and satisfies $N(x, f_b^s(\pi_r)(x)) = N(x, \pi_r(x))$ for any $x \neq s$, and $N(s, f_b^s(\pi_r)(s)) > 0$.

Then consider the modified policy $f_b^s(\pi_r)$.

If there still exists $s' \notin \mathcal{S}^0$ such that $N(s', f_b^s(\pi_r)(s')) = 0$, then similarly, there exists another action $b' \neq f_b^s(\pi_r)(s')$ such that $N(s', b') > 0$. Then whether $r(s', b') > 0$, which falls into Case (I) and leads to a contradiction, or applying Lemma 3 again implies another optimal policy $f_{b'}^{s'}(f_b^s(\pi_r))$, such that $N(s, f_{b'}^{s'}(f_b^s(\pi_r))(x)) = N(s, f_b^s(\pi_r)(x)) > 0$ for $x \notin \{s, s'\}$, $N(s, f_{b'}^{s'}(f_b^s(\pi_r))(s)) = N(s, f_b^s(\pi_r)(s)) > 0$ and $N(s', f_{b'}^{s'}(f_b^s(\pi_r))(s')) > 0$.

Repeating this procedure recursively further implies there exists an optimal policy $\pi$, such that $N(s, \pi(s)) > 0$ for any $s \notin \mathcal{S}^0$, which is a contraction to our assumption.

Therefore it completes the proof. $\qquad\square$

**Lemma 3.** *For a deterministic robust optimal policy $\pi_r$, if there exists a state $s \notin \mathcal{S}^0$ and an action $b$ such that $N(s, \pi_r(s)) = 0$, $r(s, b) = 0$ and $N(s, b) > 0$, define a modified policy $f_b^s(\pi_r)$ as*

$$f_b^s(\pi_r)(s) = b, \tag{33}$$
$$f_b^s(\pi_r)(x) = \pi_r(x), \ for \ x \neq s. \tag{34}$$

*Then the modified policy $f_b^s(\pi_r)$ is also optimal.*

*Proof.* Define a modified uncertainty set $\tilde{\mathcal{P}}_s^b$ as

$$(\tilde{\mathcal{P}}_s^b)_s^b = \Delta(\mathcal{S}), \tag{35}$$
$$(\tilde{\mathcal{P}}_s^b)_x^a = \hat{\mathcal{P}}_x^a, \ for \ (x, a) \neq (s, b). \tag{36}$$

Then we have that $\hat{\mathcal{P}}_x^a \subset (\tilde{\mathcal{P}}_s^b)_x^a$, for any $(x, a) \in \mathcal{S} \times \mathcal{A}$, which further implies that

$$V_{\hat{\mathcal{P}}}^{f_b^s(\pi_r)} \geq V_{\tilde{\mathcal{P}}_s^b}^{f_b^s(\pi_r)}. \tag{37}$$

Now we have that

$$V_{\tilde{\mathcal{P}}_s^b}^{f_b^s(\pi_r)}(s) = Q_{\tilde{\mathcal{P}}_s^b}^{f_b^s(\pi_r)}(s, b) = r(s, b) + \gamma \sigma_{(\tilde{\mathcal{P}}_s^b)_s^b}(V_{\tilde{\mathcal{P}}_s^b}^{f_b^s(\pi_r)}) \leq \gamma V_{\tilde{\mathcal{P}}_s^b}^{f_b^s(\pi_r)}(s), \tag{38}$$

which further implies $V^{f_b^s(\pi_r)}_{\tilde{\mathcal{P}}^b_s}(s) = 0$. Note that in eq. (31), we have shown $V^{\pi_r}_{\hat{\mathcal{P}}}(s) = 0$, hence $V^{f_b^s(\pi_r)}_{\tilde{\mathcal{P}}^b_s}(s) = V^{\pi_r}_{\hat{\mathcal{P}}}(s) = 0$.

Now consider the two robust Bellman operator $\mathbf{T}^s_b V(x) = \sum_a f_b^s(\pi_r)(a|x)(\hat{r}(x,a) + \gamma\sigma_{(\tilde{\mathcal{P}}^b_s)^a_x}(V))$ and $\mathbf{T}V(x) = \hat{r}(x, \pi_r(x)) + \gamma\sigma_{\hat{\mathcal{P}}^{\pi_r(x)}_x}(V)$. It is known that $V^{f_b^s(\pi_r)}_{\tilde{\mathcal{P}}^b_s}$ is the unique fixed point of the robust Bellman operator $\mathbf{T}^s_b$ and $V^{\pi_r}_{\hat{\mathcal{P}}}$ is the unique fixed point of $\mathbf{T}$.

When $x \neq s$,

$$
\begin{aligned}
\mathbf{T}^s_b V^{\pi_r}_{\hat{\mathcal{P}}}(x) &= \sum_a f_b^s(\pi_r)(a|x)(\hat{r}(x,a) + \gamma\sigma_{(\tilde{\mathcal{P}}^b_s)^a_x}(V^{\pi_r}_{\hat{\mathcal{P}}})) \\
&\overset{(a)}{=} \sum_a \pi_r(a|x)(\hat{r}(x,a) + \gamma\sigma_{(\tilde{\mathcal{P}}^b_s)^a_x}(V^{\pi_r}_{\hat{\mathcal{P}}})) \\
&= \hat{r}(x, \pi_r(x)) + \gamma\sigma_{(\tilde{\mathcal{P}}^b_s)^{\pi_r(x)}_x}(V^{\pi_r}_{\hat{\mathcal{P}}}) \\
&\overset{(b)}{=} \hat{r}(x, \pi_r(x)) + \gamma\sigma_{\hat{\mathcal{P}}^{\pi_r(x)}_x}(V^{\pi_r}_{\hat{\mathcal{P}}}) \\
&= \mathbf{T}V^{\pi_r}_{\hat{\mathcal{P}}}(x) = V^{\pi_r}_{\hat{\mathcal{P}}}(x),
\end{aligned}
\tag{39}
$$

where $(a)$ is from $f_b^s(\pi_r)(x) = \pi_r(x)$ when $x \neq s$, $(b)$ is from $(\tilde{\mathcal{P}}^b_s)^{\pi_r(x)}_x = \hat{\mathcal{P}}^{\pi_r(x)}_x$.

And for $s$, recall that $N(s, \pi_r(s)) = 0$, hence $\hat{r}(s, \pi_r(s)) = 0$ and $\hat{\mathcal{P}}^{\pi_r(s)}_s = \Delta(\mathcal{S})$. Thus

$$
\begin{aligned}
\mathbf{T}^s_b V^{\pi_r}_{\hat{\mathcal{P}}}(s) &= \hat{r}(s, b) + \gamma\sigma_{(\tilde{\mathcal{P}}^b_s)^b_s}(V^{\pi_r}_{\hat{\mathcal{P}}}) \\
&\overset{(a)}{=} \hat{r}(s, \pi_r(s)) + \gamma\sigma_{\Delta(\mathcal{S})}(V^{\pi_r}_{\hat{\mathcal{P}}}) \\
&\overset{(b)}{=} \hat{r}(s, \pi_r(s)) + \gamma\sigma_{\hat{\mathcal{P}}^{\pi_r(s)}_s}(V^{\pi_r}_{\hat{\mathcal{P}}}) \\
&= \mathbf{T}V^{\pi_r}_{\hat{\mathcal{P}}}(s) \\
&= V^{\pi_r}_{\hat{\mathcal{P}}}(s),
\end{aligned}
\tag{40}
$$

where $(a)$ is from $(\tilde{\mathcal{P}}^b_s)^b_s = \Delta(\mathcal{S})$ and $\hat{r}(s, b) = r(s, b) = 0 = \hat{r}(s, \pi_r(s))$, and $(b)$ follows from the fact $\hat{\mathcal{P}}^{\pi_r(s)}_s = \Delta(\mathcal{S})$.

Combining eq. (39) and eq. (40) further implies that $V^{\pi_r}_{\hat{\mathcal{P}}}$ is also a fixed point of $\mathbf{T}^s_b$. Hence it must be identical to $V^{f_b^s(\pi_r)}_{\tilde{\mathcal{P}}^b_s}$, i.e., $V^{f_b^s(\pi_r)}_{\tilde{\mathcal{P}}^b_s} = V^{\pi_r}_{\hat{\mathcal{P}}}$.

Combine with eq. (37), we have

$$
V^{f_b^s(\pi_r)}_{\hat{\mathcal{P}}} \geq V^{f_b^s(\pi_r)}_{\tilde{\mathcal{P}}^b_s} = V^{\pi_r}_{\hat{\mathcal{P}}},
\tag{41}
$$

which implies that $f_b^s(\pi_r)$ is also optimal with $N(s, f_b^s(\pi_r)(s)) = N(s, b) > 0$. And since $f_b^s(\pi_r)(x) = \pi_r(x)$ for $x \neq s$, then $N(x, f_b^s(\pi_r)(x)) = N(x, \pi_r(x))$. This thus completes the proof. □

**Theorem 4.** *(Restatement of Thm 2) Consider the robust MDP $\hat{\mathcal{M}}$, there exist an optimal robust policy $\pi_r$ and some universal constant $K$, such that if $N \geq \frac{1}{(1-\gamma)KSC^{\pi^*}\mu^2_{\min}}$, it holds that*

$$
V^{\pi^*}_{\mathsf{P}}(\rho) - V^{\pi_r}_{\mathsf{P}}(\rho) \leq \sqrt{\frac{KSC^{\pi^*}\log\frac{4SAN}{(1-\gamma)\delta}}{(1-\gamma)^3 N}} + \frac{384\log^2\frac{4SAN}{(1-\gamma)\delta}}{(1-\gamma)^2 N\mu_{\min}},
\tag{42}
$$

*with probability at least $1 - 4\delta$.*

*Proof.* We first have that

$$
V^{\pi^*}_{\mathsf{P}} - V^{\pi_r}_{\mathsf{P}} = \underbrace{V^{\pi^*}_{\mathsf{P}} - V^{\pi_r}_{\hat{\mathcal{P}}}}_{\Delta_1} + \underbrace{V^{\pi_r}_{\hat{\mathcal{P}}} - V^{\pi_r}_{\mathsf{P}}}_{\Delta_2}.
\tag{43}
$$

Note that equation 42 can be directly obtained if

$$N \leq \frac{SKC^{\pi^*} \log \frac{NS}{\delta}}{1-\gamma} \tag{44}$$

or

$$N \leq \frac{384 \log^2 \frac{4SAN}{(1-\gamma)\delta}}{(1-\gamma)\mu_{\min}}. \tag{45}$$

Hence in the following proof, we only focus on the case when

$$N > \max \left\{ \frac{SKC^{\pi^*} \log \frac{NS}{\delta}}{1-\gamma}, \frac{1}{(1-\gamma)KSC^{\pi^*}\mu_{\min}^2}, \frac{384 \log^2 \frac{4SAN}{(1-\gamma)\delta}}{(1-\gamma)\mu_{\min}} \right\}. \tag{46}$$

Clearly it holds that $\frac{8 \log \frac{8SA}{\delta}}{\mu_{\min}}$, and thus Lemma 8 of (Shi & Chi, 2022) states that under eq. (46), with probability $1-\delta$, it holds that

$$N(s,a) \geq \frac{N\mu(s,a)}{8 \log \frac{4SA}{\delta}}, \forall(s,a). \tag{47}$$

This moreover implies that with probability $1-\delta$, if $\mu(s,a) > 0$, then $N(s,a) > 0$. We hence focus on the case when this event holds.

The remaining proof can be completed by combining the following two theorems. □

**Theorem 5.** *With probability at least $1-2\delta$, it holds that*

$$\rho^\top \Delta_1 \leq \frac{2c_2 SC^{\pi^*} \log \frac{4SAN}{(1-\gamma)\delta}}{(1-\gamma)^2 N} + \frac{80Sc_1 C^{\pi^*} \log \frac{NS}{\delta}}{(1-\gamma)^2 N} + \frac{96}{\gamma} \sqrt{\frac{48SC^{\pi^*} \log \frac{4SAN}{(1-\gamma)\delta}}{(1-\gamma)^3 N}}. \tag{48}$$

*Proof.* We first define the following set:

$$\mathcal{S}_0 \triangleq \{s : d^{\pi^*}(s) = 0\}. \tag{49}$$

And for $s \notin \mathcal{S}_0$, it holds that $d^{\pi^*}(s) > 0$.

We first consider $s \notin \mathcal{S}_0$. Due to the fact $d^{\pi^*}(s) > 0$, hence $d^{\pi^*}(s, \pi^*(s)) = d^{\pi^*}(s)\pi^*(\pi^*(s)|s) > 0$. Thus it implies that $\mu(s, \pi^*(s)) > 0$, and equation 47 further implies $N(s, \pi^*(s)) > 0$ and $\hat{r}(s, \pi^*(s)) = r(s, \pi^*(s))$. Hence we have that

$$V_{\mathsf{P}}^{\pi^*}(s) - V_{\hat{\mathcal{P}}}^{\pi_r}(s) = r(s, \pi^*(s)) + \gamma \mathsf{P}_s^{\pi^*(s)} V_{\mathsf{P}}^{\pi^*} - V_{\hat{\mathcal{P}}}^{\pi_r}(s)$$

$$\overset{(a)}{\leq} \gamma \mathsf{P}_s^{\pi^*(s)} V_{\mathsf{P}}^{\pi^*} - \gamma \sigma_{\hat{\mathcal{P}}_s^{\pi^*(s)}}(V_{\hat{\mathcal{P}}}^{\pi_r})$$

$$= \gamma \mathsf{P}_s^{\pi^*(s)} V_{\mathsf{P}}^{\pi^*} - \gamma \mathsf{P}_s^{\pi^*(s)} V_{\hat{\mathcal{P}}}^{\pi_r} + \gamma \mathsf{P}_s^{\pi^*(s)} V_{\hat{\mathcal{P}}}^{\pi_r} - \gamma \sigma_{\hat{\mathcal{P}}_s^{\pi^*(s)}}(V_{\hat{\mathcal{P}}}^{\pi_r})$$

$$= \gamma \mathsf{P}_s^{\pi^*(s)} (V_{\mathsf{P}}^{\pi^*} - V_{\hat{\mathcal{P}}}^{\pi_r}) + \gamma (\mathsf{P}_s^{\pi^*(s)} V_{\hat{\mathcal{P}}}^{\pi_r} - \sigma_{\hat{\mathcal{P}}_s^{\pi^*(s)}}(V_{\hat{\mathcal{P}}}^{\pi_r})), \tag{50}$$

where $(a)$ is from $V_{\hat{\mathcal{P}}}^{\pi_r}(s) = \max_a Q_{\hat{\mathcal{P}}}^{\pi_r}(s, a) \geq Q_{\hat{\mathcal{P}}}^{\pi_r}(s, \pi^*(s)) = \hat{r}(s, \pi^*(s)) + \gamma \sigma_{\hat{\mathcal{P}}_s^{\pi^*(s)}}(V_{\hat{\mathcal{P}}}^{\pi_r}) = r(s, \pi^*(s)) + \gamma \sigma_{\hat{\mathcal{P}}_s^{\pi^*(s)}}(V_{\hat{\mathcal{P}}}^{\pi_r})$.

For $s \in \mathcal{S}_0$, it holds that

$$V_{\mathsf{P}}^{\pi^*}(s) - V_{\hat{\mathcal{P}}}^{\pi_r}(s) = r(s, \pi^*(s)) + \gamma \mathsf{P}_s^{\pi^*(s)} V_{\mathsf{P}}^{\pi^*} - V_{\hat{\mathcal{P}}}^{\pi_r}(s)$$

$$\overset{(a)}{\leq} \gamma \mathsf{P}_s^{\pi^*(s)} V_{\mathsf{P}}^{\pi^*} - \gamma \sigma_{\hat{\mathcal{P}}_s^{\pi^*(s)}}(V_{\hat{\mathcal{P}}}^{\pi_r}) + r(s, \pi^*(s)) - \hat{r}(s, \pi^*(s))$$

$$\leq \gamma \mathsf{P}_s^{\pi^*(s)} (V_{\mathsf{P}}^{\pi^*} - V_{\hat{\mathcal{P}}}^{\pi_r}) + \gamma (\mathsf{P}_s^{\pi^*(s)} V_{\hat{\mathcal{P}}}^{\pi_r} - \sigma_{\hat{\mathcal{P}}_s^{\pi^*(s)}}(V_{\hat{\mathcal{P}}}^{\pi_r})) + 1, \tag{51}$$

where $(a)$ follows similarly from eq. (50), and the last inequality is from $r(s, \pi^*(s)) \leq 1$.

Hence combining eq. (50) and eq. (51) implies

$$V_{\mathsf{P}}^{\pi^*}(s) - V_{\hat{\mathcal{P}}}^{\pi_r}(s) \leq \gamma \mathsf{P}_s^{\pi^*(s)}(V_{\mathsf{P}}^{\pi^*} - V_{\hat{\mathcal{P}}}^{\pi_r}) + b^*(V)(s), \tag{52}$$

where $b^*(V)(s) \triangleq \gamma \mathsf{P}_s^{\pi^*(s)} V - \gamma \sigma_{\hat{\mathcal{P}}_s^{\pi^*(s)}}(V)$ if $s \notin \mathcal{S}_0$, and $b^*(V)(s) \triangleq 1 + \gamma \mathsf{P}_s^{\pi^*(s)} V - \gamma \sigma_{\hat{\mathcal{P}}_s^{\pi^*(s)}}(V)$ for $s \in \mathcal{S}_0$.

Moreover we set

$$\tilde{b}(V_{\hat{\mathcal{P}}}^{\pi_r})(s) = \max\{0, b^*(V_{\hat{\mathcal{P}}}^{\pi_r})(s)\}, \tag{53}$$

then it holds that $b^*(V_{\hat{\mathcal{P}}}^{\pi_r}) \leq \tilde{b}(V_{\hat{\mathcal{P}}}^{\pi_r})$.

Then apply eq. (52) recursively and we have that

$$\rho^\top \Delta_1 \leq \frac{1}{1-\gamma} \left\langle d^{\pi^*}, \tilde{b}(V_{\hat{\mathcal{P}}}^{\pi_r}) \right\rangle, \tag{54}$$

Not that for $s \in \mathcal{S}_0$, it holds that $d^{\pi^*}(s) = 0$, and

$$\left\langle d^{\pi^*}, \tilde{b}(V_{\hat{\mathcal{P}}}^{\pi_r}) \right\rangle = \sum_{s \notin \mathcal{S}_0} d^{\pi^*}(s) \tilde{b}(V_{\hat{\mathcal{P}}}^{\pi_r})(s). \tag{55}$$

This implies that we only need to focus on $s \notin \mathcal{S}_0$. We further defined the following sets:

$$\mathcal{S}_s \triangleq \left\{ s \notin \mathcal{S}_0 : N\mu(s, \pi^*(s)) \leq 8 \log \frac{NS}{\delta} \right\}, \tag{56}$$

$$\mathcal{S}_l \triangleq \left\{ s \notin \mathcal{S}_0 : N\mu(s, \pi^*(s)) > 8 \log \frac{NS}{\delta} \right\}. \tag{57}$$

For $s \in \mathcal{S}_s$, we have that

$$\min \left\{ d^{\pi^*}(s), \frac{1}{S} \right\} \leq C^{\pi^*} \mu(s, \pi^*(s)) \leq \frac{8C^{\pi^*} \log \frac{NS}{\delta}}{N} \overset{(a)}{<} \frac{1}{S}, \tag{58}$$

where $(a)$ is due to the fact eq. (46).

This further implies that $d^{\pi^*}(s) \leq \frac{8C^{\pi^*} \log \frac{NS}{\delta}}{N}$. Hence

$$\sum_{s \in \mathcal{S}_s} d^{\pi^*}(s) \tilde{b}(V_{\hat{\mathcal{P}}}^{\pi_r})(s) \leq \frac{2}{1-\gamma} \frac{8SC^{\pi^*} \log \frac{NS}{\delta}}{N}, \tag{59}$$

which is due to $\|\mathsf{P}_s^{\pi^*(s)} V_{\hat{\mathcal{P}}}^{\pi_r} - \sigma_{\hat{\mathcal{P}}_s^{\pi^*(s)}}(V_{\hat{\mathcal{P}}}^{\pi_r})\| \leq \frac{2}{1-\gamma}$.

We then consider $s \in \mathcal{S}_l$. Note that from the definition and equation 47, it holds that $N(s, \pi^*(s)) > 0$ for $s \in \mathcal{S}_l$.

Therefore it holds that

$$\begin{aligned}
\tilde{b}(V_{\hat{\mathcal{P}}}^{\pi_r})(s) &\leq |\gamma \mathsf{P}_s^{\pi^*(s)} V_{\hat{\mathcal{P}}}^{\pi_r} - \gamma \sigma_{\hat{\mathcal{P}}_s^{\pi^*(s)}}(V_{\hat{\mathcal{P}}}^{\pi_r})| \\
&\leq |\gamma \mathsf{P}_s^{\pi^*(s)} V_{\hat{\mathcal{P}}}^{\pi_r} - \gamma \hat{\mathsf{P}}_s^{\pi^*(s)} V_{\hat{\mathcal{P}}}^{\pi_r}| + |\gamma \hat{\mathsf{P}}_s^{\pi^*(s)} V_{\hat{\mathcal{P}}}^{\pi_r} - \gamma \sigma_{\hat{\mathcal{P}}_s^{\pi^*(s)}}(V_{\hat{\mathcal{P}}}^{\pi_r})| \\
&\leq \gamma |\mathsf{P}_s^{\pi^*(s)} V_{\hat{\mathcal{P}}}^{\pi_r} - \hat{\mathsf{P}}_s^{\pi^*(s)} V_{\hat{\mathcal{P}}}^{\pi_r}| + \sqrt{\frac{\log \frac{4SAN}{(1-\gamma)\delta} \mathbf{Var}_{\hat{\mathsf{P}}_s^{\pi^*(s)}}(V_{\hat{\mathcal{P}}}^{\pi_r})}{N(s, \pi^*(s))}},
\end{aligned} \tag{60}$$

where the last inequality is shown as follows.

Since $\hat{\mathsf{P}}_s^{\pi^*(s)} \in \hat{\mathcal{P}}_s^a$, we have that $\hat{\mathsf{P}}_s^{\pi^*(s)} V_{\hat{\mathcal{P}}}^{\pi_r} \geq \sigma_{\hat{\mathcal{P}}_s^{\pi^*(s)}}(V_{\hat{\mathcal{P}}}^{\pi_r})$. Now note that it is shown in (Iyengar, 2005; Shi et al., 2023) that

$$\sigma_{\hat{\mathcal{P}}_s^{\pi^*(s)}}(V_{\hat{\mathcal{P}}}^{\pi_r}) = \max_{\mu \in [0, V_{\hat{\mathcal{P}}}^{\pi_r}]} \left\{ \hat{\mathsf{P}}_s^{\pi^*(s)}(V_{\hat{\mathcal{P}}}^{\pi_r} - \mu) - \sqrt{R_s^{\pi^*(s)} \mathbf{Var}_{\hat{\mathsf{P}}_s^{\pi^*(s)}}(V_{\hat{\mathcal{P}}}^{\pi_r} - \mu)} \right\}. \tag{61}$$

Thus

$$
\begin{aligned}
&\left| \gamma \hat{\mathsf{P}}_s^{\pi^*(s)} V_{\hat{\mathcal{P}}}^{\pi_r} - \gamma \sigma_{\hat{\mathcal{P}}_s^{\pi^*(s)}}(V_{\hat{\mathcal{P}}}^{\pi_r}) \right| \\
&= \gamma \hat{\mathsf{P}}_s^{\pi^*(s)} V_{\hat{\mathcal{P}}}^{\pi_r} - \gamma \sigma_{\hat{\mathcal{P}}_s^{\pi^*(s)}}(V_{\hat{\mathcal{P}}}^{\pi_r}) \\
&= \gamma \hat{\mathsf{P}}_s^{\pi^*(s)} V_{\hat{\mathcal{P}}}^{\pi_r} - \gamma \max_{\mu \in [0, V_{\hat{\mathcal{P}}}^{\pi_r}]} \left\{ \hat{\mathsf{P}}_s^{\pi^*(s)}(V_{\hat{\mathcal{P}}}^{\pi_r} - \mu) - \sqrt{R_s^{\pi^*(s)} \mathbf{Var}_{\hat{\mathsf{P}}_s^{\pi^*(s)}}(V_{\hat{\mathcal{P}}}^{\pi_r} - \mu)} \right\} \\
&\stackrel{(a)}{\leq} \gamma \hat{\mathsf{P}}_s^{\pi^*(s)} V_{\hat{\mathcal{P}}}^{\pi_r} - \gamma \left( \hat{\mathsf{P}}_s^{\pi^*(s)}(V_{\hat{\mathcal{P}}}^{\pi_r}) - \sqrt{R_s^{\pi^*(s)} \mathbf{Var}_{\hat{\mathsf{P}}_s^{\pi^*(s)}}(V_{\hat{\mathcal{P}}}^{\pi_r})} \right) \\
&= \sqrt{R_s^{\pi^*(s)} \mathbf{Var}_{\hat{\mathsf{P}}_s^{\pi^*(s)}}(V_{\hat{\mathcal{P}}}^{\pi_r})},
\end{aligned} \tag{62}
$$

where $(a)$ is due to the maximum term is larger than the function value at $\mu = 0$, and this inequality completes the proof of Equation (60).

To further bound eq. (60), we invoke Lemma 6 and have that

$$
\begin{aligned}
&\left| \mathsf{P}_s^{\pi^*(s)} V_{\hat{\mathcal{P}}}^{\pi_r} - \hat{\mathsf{P}}_s^{\pi^*(s)} V_{\hat{\mathcal{P}}}^{\pi_r} \right| \\
&\leq 12 \sqrt{\frac{\mathbf{Var}_{\hat{\mathsf{P}}_s^{\pi^*(s)}}(V_{\hat{\mathcal{P}}}^{\pi_r}) \log \frac{4SAN}{(1-\gamma)\delta}}{N(s, \pi^*(s))}} + \frac{74 \log \frac{4SAN}{(1-\gamma)\delta}}{(1-\gamma)N(s, \pi^*(s))} \\
&\leq 12 \sqrt{\frac{\log \frac{4SAN}{(1-\gamma)\delta}}{N(s, \pi^*(s))} \left( 2\mathbf{Var}_{\mathsf{P}_s^{\pi^*(s)}}(V_{\hat{\mathcal{P}}}^{\pi_r}) + \frac{41 \log \frac{4SAN}{(1-\gamma)\delta}}{(1-\gamma)^2 N(s, \pi^*(s))} \right)} + \frac{74 \log \frac{4SAN}{(1-\gamma)\delta}}{(1-\gamma)N(s, \pi^*(s))} \\
&\leq 12 \sqrt{\frac{2 \log \frac{4SAN}{(1-\gamma)\delta}}{N(s, \pi^*(s))} \mathbf{Var}_{\mathsf{P}_s^{\pi^*(s)}}(V_{\hat{\mathcal{P}}}^{\pi_r})} + \frac{(74 + 12\sqrt{41}) \log \frac{4SAN}{(1-\gamma)\delta}}{(1-\gamma)N(s, \pi^*(s))},
\end{aligned} \tag{63}
$$

where the last inequality is from $\sqrt{x+y} \leq \sqrt{x} + \sqrt{y}$.

Combine eq. (60) and eq. (63), we further have that

$$
\begin{aligned}
\tilde{b}(V_{\hat{\mathcal{P}}}^{\pi_r})(s) &\leq 24 \sqrt{\frac{2 \log \frac{4SAN}{(1-\gamma)\delta}}{N(s, \pi^*(s))} \mathbf{Var}_{\mathsf{P}_s^{\pi^*(s)}}(V_{\hat{\mathcal{P}}}^{\pi_r})} + \frac{(74 + 12\sqrt{41}) \log \frac{4SAN}{(1-\gamma)\delta}}{(1-\gamma)N(s, \pi^*(s))} + \frac{\log \frac{N}{\delta}}{(1-\gamma)N(s, \pi^*(s))} \\
&\leq 24 \sqrt{\frac{2 \log \frac{4SAN}{(1-\gamma)\delta}}{N(s, \pi^*(s))} \mathbf{Var}_{\mathsf{P}_s^{\pi^*(s)}}(V_{\hat{\mathcal{P}}}^{\pi_r})} + \frac{c_1 \log \frac{4SAN}{(1-\gamma)\delta}}{(1-\gamma)N(s, \pi^*(s))},
\end{aligned} \tag{64}
$$

where $c_1 = 75 + 12\sqrt{41}$.

Note that in eq. (25), we showed that $\frac{1}{N(s, \pi^*(s))} \leq \frac{12C^{\pi^*}}{N}(S + \frac{1}{d^{\pi^*}(s)})$. Hence plugging in eq. (64) implies that

$$
\begin{aligned}
\tilde{b}(V_{\hat{\mathcal{P}}}^{\pi_r})(s) &\leq 24 \sqrt{\frac{24 C^{\pi^*} \log \frac{4SAN}{(1-\gamma)\delta}}{N} \mathbf{Var}_{\mathsf{P}_s^{\pi^*(s)}}(V_{\hat{\mathcal{P}}}^{\pi_r})} \left( \sqrt{S} + \frac{1}{\sqrt{d^{\pi^*}(s)}} \right) \\
&+ \frac{12 c_1 C^{\pi^*} \log \frac{4SAN}{(1-\gamma)\delta}}{(1-\gamma)N} \left( S + \frac{1}{d^{\pi^*}(s)} \right).
\end{aligned} \tag{65}
$$

Firstly we have that

$$\sum_{s \in \mathcal{S}_l} 24 d^{\pi^*}(s) \sqrt{\frac{24 C^{\pi^*} \log \frac{4SAN}{(1-\gamma)\delta}}{N} \mathbf{Var}_{\mathsf{P}_s^{\pi^*(s)}}(V_{\hat{\mathcal{P}}}^{\pi_r})} \left( \sqrt{S} + \frac{1}{\sqrt{d^{\pi^*}(s)}} \right)$$

$$= \sum_{s \in \mathcal{S}_l} 24 d^{\pi^*}(s) \sqrt{\frac{24 S C^{\pi^*} \log \frac{4SAN}{(1-\gamma)\delta}}{N} \mathbf{Var}_{\mathsf{P}_s^{\pi^*(s)}}(V_{\hat{\mathcal{P}}}^{\pi_r})} + \sum_{s \in \mathcal{S}_l} 12 \sqrt{d^{\pi^*}(s)} \sqrt{\frac{24 C^{\pi^*} \log \frac{4SAN}{(1-\gamma)\delta}}{N} \mathbf{Var}_{\mathsf{P}_s^{\pi^*(s)}}(V_{\hat{\mathcal{P}}}^{\pi_r})}$$

$$= 24 \sqrt{\frac{24 C^{\pi^*} \log \frac{4SAN}{(1-\gamma)\delta}}{N}} \left( \sum_{s \in \mathcal{S}_l} \sqrt{d^{\pi^*}(s) \mathbf{Var}_{\mathsf{P}_s^{\pi^*(s)}}(V_{\hat{\mathcal{P}}}^{\pi_r})} + \sum_{s \in \mathcal{S}_l} \sqrt{d^{\pi^*}(s)} \sqrt{S d^{\pi^*}(s) \mathbf{Var}_{\mathsf{P}_s^{\pi^*(s)}}(V_{\hat{\mathcal{P}}}^{\pi_r})} \right)$$

$$\overset{(a)}{\leq} 24 \sqrt{\frac{24 C^{\pi^*} \log \frac{4SAN}{(1-\gamma)\delta}}{N}} \left( \sqrt{S} \sqrt{\sum_{s \in \mathcal{S}_l} d^{\pi^*}(s) \mathbf{Var}_{\mathsf{P}_s^{\pi^*(s)}}(V_{\hat{\mathcal{P}}}^{\pi_r})} + \sqrt{\sum_{s \in \mathcal{S}_l} S d^{\pi^*}(s) \mathbf{Var}_{\mathsf{P}_s^{\pi^*(s)}}(V_{\hat{\mathcal{P}}}^{\pi_r})} \right)$$

$$= 48 \sqrt{\frac{24 S C^{\pi^*} \log \frac{4SAN}{(1-\gamma)\delta}}{N}} \sqrt{\sum_{s \in \mathcal{S}_l} d^{\pi^*}(s) \mathbf{Var}_{\mathsf{P}_s^{\pi^*(s)}}(V_{\hat{\mathcal{P}}}^{\pi_r})}, \tag{66}$$

where $(a)$ is from Cauchy's inequality and the fact $\sum_{s \in \mathcal{S}_l} d^{\pi^*}(s) \leq 1$.

In addition, we have that

$$\sum_{s \in \mathcal{S}_l} d^{\pi^*}(s) \frac{12 c_1 C^{\pi^*} \log \frac{4SAN}{(1-\gamma)\delta}}{(1-\gamma)N} \left( S + \frac{1}{d^{\pi^*}(s)} \right) \leq \frac{24 c_1 S C^{\pi^*} \log \frac{4SAN}{(1-\gamma)\delta}}{(1-\gamma)N}. \tag{67}$$

Combine the two inequalities above and we have that

$$\sum_{s \in \mathcal{S}_l} d^{\pi^*}(s) \tilde{b}(V_{\hat{\mathcal{P}}}^{\pi_r})(s)$$

$$\leq 48 \sqrt{\frac{24 S C^{\pi^*} \log \frac{4SAN}{(1-\gamma)\delta}}{N}} \sqrt{\sum_{s \in \mathcal{S}_l} d^{\pi^*}(s) \mathbf{Var}_{\mathsf{P}_s^{\pi^*(s)}}(V_{\hat{\mathcal{P}}}^{\pi_r})} + \frac{24 S c_1 C^{\pi^*} \log \frac{4SAN}{(1-\gamma)\delta}}{(1-\gamma)N}. \tag{68}$$

Then we combine eq. (59) and eq. (68), and it implies that

$$\langle d^{\pi^*}, \tilde{b}(V_{\hat{\mathcal{P}}}^{\pi_r}) \rangle$$

$$= \sum_{s \in \mathcal{S}_s} d^{\pi^*}(s) \tilde{b}(V_{\hat{\mathcal{P}}}^{\pi_r})(s) + \sum_{s \in \mathcal{S}_l} d^{\pi^*}(s) \tilde{b}(V_{\hat{\mathcal{P}}}^{\pi_r})(s)$$

$$\leq \frac{16 S C^{\pi^*} \log \frac{NS}{\delta}}{(1-\gamma)N} + 48 \sqrt{\frac{24 S C^{\pi^*} \log \frac{4SAN}{(1-\gamma)\delta}}{N}} \sqrt{\sum_{s \in \mathcal{S}_l} d^{\pi^*}(s) \mathbf{Var}_{\mathsf{P}_s^{\pi^*(s)}}(V_{\hat{\mathcal{P}}}^{\pi_r})} + \frac{24 S c_1 C^{\pi^*} \log \frac{4SAN}{(1-\gamma)\delta}}{(1-\gamma)N}$$

$$\leq \frac{40 c_1 S C^{\pi^*} \log \frac{NS}{\delta}}{(1-\gamma)N} + 48 \sqrt{\frac{24 S C^{\pi^*} \log \frac{4SAN}{(1-\gamma)\delta}}{N}} \sqrt{\sum_{s \in \mathcal{S}} d^{\pi^*}(s) \mathbf{Var}_{\mathsf{P}_s^{\pi^*(s)}}(V_{\hat{\mathcal{P}}}^{\pi_r})}. \tag{69}$$

We then bound the term $\sum_{s \in \mathcal{S}} d^{\pi^*}(s) \mathbf{Var}_{\mathsf{P}_s^{\pi^*(s)}}(V_{\hat{\mathcal{P}}}^{\pi_r})$. We first claim the following inequality:

$$V_{\hat{\mathcal{P}}}^{\pi_r} - \gamma \mathsf{P}^{\pi^*} V_{\hat{\mathcal{P}}}^{\pi_r} + 2\tilde{b}(V_{\hat{\mathcal{P}}}^{\pi_r}) \geq 0. \tag{70}$$

To prove eq. (70), we note that

$$V_{\hat{\mathcal{P}}}^{\pi_r}(s) = \max_a Q_{\hat{\mathcal{P}}}^{\pi_r}(s, a)$$

$$\geq Q_{\hat{\mathcal{P}}}^{\pi_r}(s, \pi^*(s))$$

$$= \hat{r}(s, \pi^*(s)) + \gamma\sigma_{\hat{\mathcal{P}}_s^{\pi^*(s)}}(V_{\hat{\mathcal{P}}}^{\pi_r})$$

$$= \hat{r}(s, \pi^*(s)) + \gamma\mathsf{P}_s^{\pi^*(s)}V_{\hat{\mathcal{P}}}^{\pi_r} - \gamma\mathsf{P}_s^{\pi^*(s)}V_{\hat{\mathcal{P}}}^{\pi_r} + \gamma\sigma_{\hat{\mathcal{P}}_s^{\pi^*(s)}}(V_{\hat{\mathcal{P}}}^{\pi_r})$$

$$\overset{(a)}{\geq} \hat{r}(s, \pi^*(s)) + \gamma\mathsf{P}_s^{\pi^*(s)}V_{\hat{\mathcal{P}}}^{\pi_r} - b^*(V_{\hat{\mathcal{P}}}^{\pi_r})(s)$$

$$\geq \hat{r}(s, \pi^*(s)) + \gamma\mathsf{P}_s^{\pi^*(s)}V_{\hat{\mathcal{P}}}^{\pi_r} - 2\tilde{b}(V_{\hat{\mathcal{P}}}^{\pi_r})(s), \tag{71}$$

where $(a)$ is from $b^*(V_{\hat{\mathcal{P}}}^{\pi_r})(s) \geq \gamma\mathsf{P}_s^{\pi^*(s)}V_{\hat{\mathcal{P}}}^{\pi_r} - \gamma\sigma_{\hat{\mathcal{P}}_s^{\pi^*(s)}}(V_{\hat{\mathcal{P}}}^{\pi_r})$.

Hence for any $s \in \mathcal{S}$,

$$V_{\hat{\mathcal{P}}}^{\pi_r}(s) - \gamma\mathsf{P}_s^{\pi^*(s)}V_{\hat{\mathcal{P}}}^{\pi_r} + 2\tilde{b}(V_{\hat{\mathcal{P}}}^{\pi_r})(s) \geq \hat{r}(s, \pi^*(s)) \geq 0, \tag{72}$$

which proves eq. (70).

Now with eq. (70), we first note that

$$(V_{\hat{\mathcal{P}}}^{\pi_r} \circ V_{\hat{\mathcal{P}}}^{\pi_r}) - (\gamma\mathsf{P}^{\pi^*}V_{\hat{\mathcal{P}}}^{\pi_r}) \circ (\gamma\mathsf{P}^{\pi^*}V_{\hat{\mathcal{P}}}^{\pi_r})$$

$$= (V_{\hat{\mathcal{P}}}^{\pi_r} - \gamma\mathsf{P}^{\pi^*}V_{\hat{\mathcal{P}}}^{\pi_r}) \circ (V_{\hat{\mathcal{P}}}^{\pi_r} + \gamma\mathsf{P}^{\pi^*}V_{\hat{\mathcal{P}}}^{\pi_r})$$

$$\leq (V_{\hat{\mathcal{P}}}^{\pi_r} - \gamma\mathsf{P}^{\pi^*}V_{\hat{\mathcal{P}}}^{\pi_r} + 2\tilde{b}(V_{\hat{\mathcal{P}}}^{\pi_r})) \circ (V_{\hat{\mathcal{P}}}^{\pi_r} + \gamma\mathsf{P}^{\pi^*}V_{\hat{\mathcal{P}}}^{\pi_r})$$

$$\leq \frac{2}{1-\gamma}(V_{\hat{\mathcal{P}}}^{\pi_r} - \gamma\mathsf{P}^{\pi^*}V_{\hat{\mathcal{P}}}^{\pi_r} + 2\tilde{b}(V_{\hat{\mathcal{P}}}^{\pi_r})), \tag{73}$$

where the last inequality is due to the fact $\|V_{\hat{\mathcal{P}}}^{\pi_r} + \gamma\mathsf{P}^{\pi^*}V_{\hat{\mathcal{P}}}^{\pi_r}\| \leq \frac{2}{1-\gamma}$ and eq. (70).

We then have that

$$\sum_{s \in \mathcal{S}} d^{\pi^*}(s)\mathbf{Var}_{\mathsf{P}_s^{\pi^*(s)}}(V_{\hat{\mathcal{P}}}^{\pi_r})$$

$$= \langle d^{\pi^*}, \mathsf{P}^{\pi^*}(V_{\hat{\mathcal{P}}}^{\pi_r} \circ V_{\hat{\mathcal{P}}}^{\pi_r}) - (\mathsf{P}^{\pi^*}V_{\hat{\mathcal{P}}}^{\pi_r}) \circ (\mathsf{P}^{\pi^*}V_{\hat{\mathcal{P}}}^{\pi_r})\rangle$$

$$\overset{(a)}{\leq} \left\langle d^{\pi^*}, \mathsf{P}^{\pi^*}(V_{\hat{\mathcal{P}}}^{\pi_r} \circ V_{\hat{\mathcal{P}}}^{\pi_r}) - \frac{1}{\gamma^2}(V_{\hat{\mathcal{P}}}^{\pi_r} \circ V_{\hat{\mathcal{P}}}^{\pi_r}) + \frac{2}{\gamma^2(1-\gamma)}(V_{\hat{\mathcal{P}}}^{\pi_r} - \gamma\mathsf{P}^{\pi^*}V_{\hat{\mathcal{P}}}^{\pi_r} + 2\tilde{b}(V_{\hat{\mathcal{P}}}^{\pi_r}))\right\rangle$$

$$\overset{(b)}{\leq} \left\langle d^{\pi^*}, \mathsf{P}^{\pi^*}(V_{\hat{\mathcal{P}}}^{\pi_r} \circ V_{\hat{\mathcal{P}}}^{\pi_r}) - \frac{1}{\gamma}(V_{\hat{\mathcal{P}}}^{\pi_r} \circ V_{\hat{\mathcal{P}}}^{\pi_r}) + \frac{2}{\gamma^2(1-\gamma)}(I - \gamma\mathsf{P}^{\pi^*})V_{\hat{\mathcal{P}}}^{\pi_r} + \frac{4}{\gamma^2(1-\gamma)}\tilde{b}(V_{\hat{\mathcal{P}}}^{\pi_r}))\right\rangle$$

$$= \left\langle d^{\pi^*}, \frac{1}{\gamma}(\gamma\mathsf{P}^{\pi^*} - I)(V_{\hat{\mathcal{P}}}^{\pi_r} \circ V_{\hat{\mathcal{P}}}^{\pi_r}) + \frac{2}{\gamma^2(1-\gamma)}(I - \gamma\mathsf{P}^{\pi^*})V_{\hat{\mathcal{P}}}^{\pi_r} + \frac{4}{\gamma^2(1-\gamma)}\tilde{b}(V_{\hat{\mathcal{P}}}^{\pi_r}))\right\rangle$$

$$= (d^{\pi^*})^\top(I - \gamma\mathsf{P}^{\pi^*})\left(-\frac{1}{\gamma}(V_{\hat{\mathcal{P}}}^{\pi_r} \circ V_{\hat{\mathcal{P}}}^{\pi_r}) + \frac{2}{\gamma^2(1-\gamma)}V_{\hat{\mathcal{P}}}^{\pi_r}\right) + \frac{4}{\gamma^2(1-\gamma)}\langle d^{\pi^*}, \tilde{b}(V_{\hat{\mathcal{P}}}^{\pi_r})\rangle$$

$$\overset{(c)}{=} (1-\gamma)\rho^\top\left(-\frac{1}{\gamma}(V_{\hat{\mathcal{P}}}^{\pi_r} \circ V_{\hat{\mathcal{P}}}^{\pi_r}) + \frac{2}{\gamma^2(1-\gamma)}V_{\hat{\mathcal{P}}}^{\pi_r}\right) + \frac{4}{\gamma^2(1-\gamma)}\langle d^{\pi^*}, \tilde{b}(V_{\hat{\mathcal{P}}}^{\pi_r})\rangle$$

$$\leq \frac{2}{\gamma^2}\rho^\top V_{\hat{\mathcal{P}}}^{\pi_r} + \frac{4}{\gamma^2(1-\gamma)}\langle d^{\pi^*}, \tilde{b}(V_{\hat{\mathcal{P}}}^{\pi_r})\rangle$$

$$\leq \frac{2}{\gamma^2(1-\gamma)} + \frac{4}{\gamma^2(1-\gamma)}\langle d^{\pi^*}, \tilde{b}(V_{\hat{\mathcal{P}}}^{\pi_r})\rangle, \tag{74}$$

where $(a)$ is from eq. (73), $(b)$ is due to $\gamma < 1$, $(c)$ is from the definition of visitation distribution.

Hence by plugging eq. (74) in eq. (69), we have that

$$\langle d^{\pi^*}, \tilde{b}(V_{\hat{\mathcal{P}}}^{\pi_r})\rangle$$

$$\leq \frac{40c_1 SC^{\pi^*}\log\frac{NS}{\delta}}{(1-\gamma)N} + 48\sqrt{\frac{24SC^{\pi^*}\log\frac{4SAN}{(1-\gamma)\delta}}{N}}\sqrt{\sum_{s \in \mathcal{S}} d^{\pi^*}(s)\mathbf{Var}_{\mathsf{P}_s^{\pi^*(s)}}(V_{\hat{\mathcal{P}}}^{\pi_r})}$$

$$\leq \frac{40c_1 SC^{\pi^*}\log\frac{NS}{\delta}}{(1-\gamma)N} + 48\sqrt{\frac{24SC^{\pi^*}\log\frac{4SAN}{(1-\gamma)\delta}}{N}}\sqrt{\frac{2}{\gamma^2(1-\gamma)} + \frac{4}{\gamma^2(1-\gamma)}}\langle d^{\pi^*}, \tilde{b}(V_{\hat{\mathcal{P}}}^{\pi_r})\rangle$$

$$\leq \frac{40c_1 SC^{\pi^*}\log\frac{NS}{\delta}}{(1-\gamma)N} + \frac{24}{\gamma}\sqrt{\frac{48SC^{\pi^*}\log\frac{4SAN}{(1-\gamma)\delta}}{(1-\gamma)N}} + \frac{48}{\gamma}\sqrt{\frac{96SC^{\pi^*}\log\frac{4SAN}{(1-\gamma)\delta}}{(1-\gamma)N}}\sqrt{\langle d^{\pi^*}, \tilde{b}(V_{\hat{\mathcal{P}}}^{\pi_r})\rangle}$$

$$\stackrel{(a)}{\leq} \frac{1}{2}\langle d^{\pi^*}, \tilde{b}(V_{\hat{\mathcal{P}}}^{\pi_r})\rangle + c_2\frac{SC^{\pi^*}\log\frac{4SAN}{(1-\gamma)\delta}}{(1-\gamma)N} + \frac{40Sc_1 C^{\pi^*}\log\frac{NS}{\delta}}{(1-\gamma)N} + \frac{48}{\gamma}\sqrt{\frac{48SC^{\pi^*}\log\frac{4SAN}{(1-\gamma)\delta}}{(1-\gamma)N}}, \tag{75}$$

where $(a)$ is from $x + y \geq 2\sqrt{xy}$ and $c_2 = 8 * 24^3 = 110592$. This inequality moreover implies that

$$\langle d^{\pi^*}, \tilde{b}(V_{\hat{\mathcal{P}}}^{\pi_r})\rangle \leq 2c_2\frac{SC^{\pi^*}\log\frac{4SAN}{(1-\gamma)\delta}}{(1-\gamma)N} + \frac{80Sc_1 C^{\pi^*}\log\frac{NS}{\delta}}{(1-\gamma)N} + \frac{96}{\gamma}\sqrt{\frac{48SC^{\pi^*}\log\frac{4SAN}{(1-\gamma)\delta}}{(1-\gamma)N}}. \tag{76}$$

Recall the definition of $\Delta_1$, we hence have that

$$\rho^\top\Delta_1 \leq \frac{2c_2 SC^{\pi^*}\log\frac{4SAN}{(1-\gamma)\delta}}{(1-\gamma)^2 N} + \frac{80Sc_1 C^{\pi^*}\log\frac{NS}{\delta}}{(1-\gamma)^2 N} + \frac{96}{\gamma}\sqrt{\frac{48SC^{\pi^*}\log\frac{4SAN}{(1-\gamma)\delta}}{(1-\gamma)^3 N}}. \tag{77}$$

This hence completes the proof of the lemma. $\square$

**Theorem 6.** *With probability at least $1 - 2\delta$, it holds that*

$$\rho^\top\Delta_2 \leq 2\sqrt{\frac{384\log^2\frac{4SAN}{(1-\gamma)\delta}}{(1-\gamma)^3 N^2\mu_{\min}}} + 2\sqrt{\frac{384\log^2\frac{4SAN}{(1-\gamma)\delta}}{(1-\gamma)^3 N^3\mu_{\min}}} + \frac{384\log^2\frac{4SAN}{(1-\gamma)\delta}}{(1-\gamma)^2 N\mu_{\min}}. \tag{78}$$

*Proof.* We first define the following set:

$$\mathcal{S}^0 \triangleq \{s \in \mathcal{S} : N(s) = 0\}. \tag{79}$$

Note that $N(s) = \sum_a N(s, a)$, hence it holds that $N(s, a) = 0$ for any $s \in \mathcal{S}^0$, $a \in \mathcal{A}$.

We moreover construct an absorbing MDP $\bar{M} = (\mathcal{S}, \mathcal{A}, \hat{r}, \bar{\mathsf{P}})$ as follows. For $s \in \mathcal{S}^0$, $\bar{\mathsf{P}}_{s,x}^a = \mathbf{1}_{x=s}$; And for $s \notin \mathcal{S}^0$, set $\bar{\mathsf{P}}_{s,x}^a = \mathsf{P}_{s,x}^a$.

Then for any $s \in \mathcal{S}^0$, from Lemma 2, it holds that $V_{\hat{\mathcal{P}}}^{\pi_r}(s) = 0$, and hence

$$V_{\hat{\mathcal{P}}}^{\pi_r}(s) - V_{\mathsf{P}}^{\pi_r}(s) \leq 0. \tag{80}$$

It further implies that

$$V_{\hat{\mathcal{P}}}^{\pi_r}(s) - V_{\mathsf{P}}^{\pi_r}(s) = \bar{\mathsf{P}}_s^{\pi_r(s)}(V_{\hat{\mathcal{P}}}^{\pi_r} - V_{\mathsf{P}}^{\pi_r}) \leq \gamma\bar{\mathsf{P}}_s^{\pi_r(s)}(V_{\hat{\mathcal{P}}}^{\pi_r} - V_{\mathsf{P}}^{\pi_r}). \tag{81}$$

On the other hand, for $s \notin \mathcal{S}^0$, Lemma 2 implies that $N(s, \pi_r(s)) > 0$, hence equation 47 implies $N(s, \pi_r(s)) > 0$, $\hat{r}(s, \pi_r(s)) = r(s, \pi_r(s))$. Hence

$$\begin{aligned}
V_{\hat{\mathcal{P}}}^{\pi_r}(s) - V_{\mathsf{P}}^{\pi_r}(s) &\stackrel{(a)}{=} \gamma\sigma_{\hat{\mathcal{P}}_s^{\pi_r(s)}}(V_{\hat{\mathcal{P}}}^{\pi_r}) - \gamma\mathsf{P}_s^{\pi_r(s)}V_{\mathsf{P}}^{\pi_r} \\
&= \gamma(\sigma_{\hat{\mathcal{P}}_s^{\pi_r(s)}}(V_{\hat{\mathcal{P}}}^{\pi_r}) - \mathsf{P}_s^{\pi_r(s)}V_{\hat{\mathcal{P}}}^{\pi_r} + \mathsf{P}_s^{\pi_r(s)}V_{\hat{\mathcal{P}}}^{\pi_r} - \mathsf{P}_s^{\pi_r(s)}V_{\mathsf{P}}^{\pi_r}) \\
&= \gamma\mathsf{P}_s^{\pi_r(s)}(V_{\hat{\mathcal{P}}}^{\pi_r} - V_{\mathsf{P}}^{\pi_r}) + \gamma(\sigma_{\hat{\mathcal{P}}_s^{\pi_r(s)}}(V_{\hat{\mathcal{P}}}^{\pi_r}) - \mathsf{P}_s^{\pi_r(s)}V_{\hat{\mathcal{P}}}^{\pi_r}) \\
&\triangleq \gamma\bar{\mathsf{P}}_s^{\pi_r(s)}(V_{\hat{\mathcal{P}}}^{\pi_r} - V_{\mathsf{P}}^{\pi_r}) + c(s),
\end{aligned} \tag{82}$$

where $(a)$ is from $r(s, \pi_r(s)) = \hat{r}(s, \pi_r(s))$, and $c(s) \triangleq \gamma(\sigma_{\hat{\mathcal{P}}_s^{\pi_r(s)}}(V_{\hat{\mathcal{P}}}^{\pi_r}) - \mathsf{P}_s^{\pi_r(s)}V_{\hat{\mathcal{P}}}^{\pi_r})$.

According to the bound we obtained in Lemma 7, it holds that

$$c(s) \leq 2\sqrt{\frac{48 \log \frac{4SAN}{(1-\gamma)\delta}\epsilon_1}{(1-\gamma)N(s,\pi_r(s))}} + 2\epsilon_1\sqrt{\frac{48 \log \frac{4SAN}{(1-\gamma)\delta}}{N(s,\pi_r(s))}} + \frac{48 \log \frac{4SAN}{(1-\gamma)\delta}}{(1-\gamma)N(s,\pi_r(s))}. \tag{83}$$

Combine eq. (81) and eq. (82), then

$$V_{\hat{\mathcal{P}}}^{\pi_r}(s) - V_{\mathsf{P}}^{\pi_r}(s) \leq \gamma \bar{\mathsf{P}}_s^{\pi_r(s)}(V_{\hat{\mathcal{P}}}^{\pi_r} - V_{\mathsf{P}}^{\pi_r}) + \tilde{c}(s), \tag{84}$$

where

$$\tilde{c}(s) = \begin{cases} 2\sqrt{\frac{48 \log \frac{4SAN}{(1-\gamma)\delta}\epsilon_1}{(1-\gamma)N(s,\pi_r(s))}} + 2\epsilon_1\sqrt{\frac{48 \log \frac{4SAN}{(1-\gamma)\delta}}{N(s,\pi_r(s))}} + \frac{48 \log \frac{4SAN}{(1-\gamma)\delta}}{(1-\gamma)N(s,\pi_r(s))}, & s \notin \mathcal{S}^0 \\ 0, & s \in \mathcal{S}^0 \end{cases} \tag{85}$$

Applying eq. (84) recursively further implies

$$\rho^\top \Delta_2 \leq \frac{1}{1-\gamma}\langle \bar{d}^{\pi_r}, \tilde{c}\rangle, \tag{86}$$

where $\bar{d}^{\pi_r}$ is the discounted visitation distribution induced by $\pi_r$ and $\bar{\mathsf{P}}$.

Note that eq. (46) and Lemma 8 of (Shi & Chi, 2022) state that with probability $1 - \delta$, for any $(s,a)$ pair,

$$N(s,a) \geq \frac{N\mu(s,a)}{8 \log \frac{4SA}{\delta}}. \tag{87}$$

Hence under this event, $\tilde{c}(s)$ can be bounded as

$$\tilde{c}(s) \leq 2\sqrt{\frac{384 \log^2 \frac{4SAN}{(1-\gamma)\delta}\epsilon_1}{(1-\gamma)N\mu_{\min}}} + 2\epsilon_1\sqrt{\frac{384 \log^2 \frac{4SAN}{(1-\gamma)\delta}}{N\mu_{\min}}} + \frac{384 \log^2 \frac{4SAN}{(1-\gamma)\delta}}{(1-\gamma)N\mu_{\min}}. \tag{88}$$

Hence we have that

$$\begin{aligned} \rho^\top \Delta_2 &\leq \frac{1}{1-\gamma}\langle \bar{d}^{\pi_r}, \tilde{c}\rangle \\ &\leq 2\sqrt{\frac{384 \log^2 \frac{4SAN}{(1-\gamma)\delta}\epsilon_1}{(1-\gamma)^3 N\mu_{\min}}} + 2\epsilon_1\sqrt{\frac{384 \log^2 \frac{4SAN}{(1-\gamma)\delta}}{(1-\gamma)^3 N\mu_{\min}}} + \frac{384 \log^2 \frac{4SAN}{(1-\gamma)\delta}}{(1-\gamma)^2 N\mu_{\min}} \\ &\leq 2\sqrt{\frac{384 \log^2 \frac{4SAN}{(1-\gamma)\delta}}{(1-\gamma)^3 N^2\mu_{\min}}} + 2\sqrt{\frac{384 \log^2 \frac{4SAN}{(1-\gamma)\delta}}{(1-\gamma)^3 N^3\mu_{\min}}} + \frac{384 \log^2 \frac{4SAN}{(1-\gamma)\delta}}{(1-\gamma)^2 N\mu_{\min}}. \end{aligned} \tag{89}$$

$$\square$$

## D  Auxiliary Lemmas

**Lemma 4** (Lemma 4, (Li et al., 2022)). *For any $\delta$, with probability $1-\delta$, $\max\{12N(s,a), 8\log\frac{NS}{\delta}\} \geq N\mu(s,a)$, $\forall s, a$.*

**Lemma 5** (Lemma 9, (Li et al., 2022)). *For any $(s,a)$ pair with $N(s,a) > 0$, if $V$ is an vector independent of $\hat{\mathsf{P}}_s^a$ obeying $\|V\| \leq \frac{1}{1-\gamma}$, then with probability at least $1 - \delta$,*

$$|(\hat{\mathsf{P}}_s^a - \mathsf{P}_s^a)V| \leq \sqrt{\frac{48 \mathbf{Var}_{\hat{\mathsf{P}}_s^a}(V) \log \frac{4N}{\delta}}{N(s,a)}} + \frac{48 \log \frac{4N}{\delta}}{(1-\gamma)N(s,a)}, \tag{90}$$

$$\mathbf{Var}_{\hat{\mathsf{P}}_s^a}(V) \leq 2\mathbf{Var}_{\mathsf{P}_s^a}(V) + \frac{5 \log \frac{4N}{\delta}}{3(1-\gamma)^2 N(s,a)}. \tag{91}$$

**Lemma 6.** *Suppose $\gamma \in [0.5, 1)$, with probability at least $1 - \delta$, it holds*

$$|(\hat{\mathsf{P}}_s^a - \mathsf{P}_s^a)V_{\hat{\mathcal{P}}}^{\pi_r}| \leq 12\sqrt{\frac{\mathbf{Var}_{\hat{\mathsf{P}}_s^a}(V_{\hat{\mathcal{P}}}^{\pi_r})\log\frac{4SAN}{(1-\gamma)\delta}}{N(s,a)}} + \frac{74\log\frac{4SAN}{(1-\gamma)\delta}}{(1-\gamma)N(s,a)}, \tag{92}$$

$$\mathbf{Var}_{\hat{\mathsf{P}}_s^a}(V_{\hat{\mathcal{P}}}^{\pi_r}) \leq 2\,\mathbf{Var}_{\mathsf{P}_s^a}(V_{\hat{\mathcal{P}}}^{\pi_r}) + \frac{41\log\frac{4SAN}{(1-\gamma)\delta}}{(1-\gamma)^2N(s,a)}. \tag{93}$$

*simultaneously for any pair $(s, a) \in \mathcal{S} \times \mathcal{A}$.*

*Proof.* When $N(s, a) = 0$, the results hold naturally. We hence only consider $(s, a)$ with $N(s, a) > 0$. **Part 1**.

Recall that $\hat{\mathsf{M}} = (\mathcal{S}, \mathcal{A}, \hat{\mathsf{P}}, \hat{r})$ is the estimated MDP. For any state $s$ and positive scalar $u > 0$, we first construct an auxiliary state-absorbing MDP $\hat{\mathsf{M}}^{s,u} = (\mathcal{S}, \mathcal{A}, \mathsf{P}^{s,u}, r^{s,u})$ as follows.

For all states except $s$, the MDP structure of $\hat{\mathsf{M}}^{s,u}$ is identical to $\hat{\mathsf{M}}$, i.e., for any $x \neq s$ and $a \in \mathcal{A}$,

$$(\mathsf{P}^{s,u})_{x,\cdot}^a = \hat{\mathsf{P}}_{x,\cdot}^a, r^{s,u}(x,a) = \hat{r}(x,a); \tag{94}$$

State $s$ is an absorbing state in $\hat{\mathsf{M}}^{s,u}$, namely, for any $a \in \mathcal{A}$,

$$(\mathsf{P}^{s,u})_{s,x}^a = \mathbf{1}_{x=s}, r^{s,u}(s,a) = u. \tag{95}$$

We then define a robust MDP $\mathcal{M}^{s,u} = (\mathcal{S}, \mathcal{A}, \mathcal{P}^{s,u}, r^{s,u})$ centered at $\hat{\mathsf{M}}^{s,u}$ as following: the uncertainty set $\mathcal{P}^{s,u}$ is defined as $\hat{\mathcal{P}} = \bigotimes_{x,a}(\mathcal{P}^{s,u})_x^a$, where if $x \neq s$,

$$(\mathcal{P}^{s,u})_x^a = \{q \in \Delta(\mathcal{S}) : D(q, (\mathsf{P}^{s,u})_x^a) \leq R_x^a\} \tag{96}$$

and

$$(\mathcal{P}^{s,u})_s^a = \{\mathbf{1}_s\}. \tag{97}$$

The optimal robust value function of $\mathcal{M}^{s,u}$ is denoted by $V^{s,u}$.

**Part 2**. We claim that if we choose $u^* = (1 - \gamma)V_{\hat{\mathcal{P}}}^{\pi_r}(s)$, then $V^{s,u^*} = V_{\hat{\mathcal{P}}}^{\pi_r}$. We prove this as follows.

Firstly note that the function $V^{s,u^*}$ is the unique fixed point of the operator $\mathbf{T}^{s,u^*}(V)(x) = \max_a\{r^{s,u^*}(x,a) + \gamma\sigma_{(\mathcal{P}^{s,u^*})_x^a}(V)\}$.

For $x \neq s$, we note that

$$\begin{aligned}
\mathbf{T}^{s,u^*}(V_{\hat{\mathcal{P}}}^{\pi_r})(x) &= \max_a\{r^{s,u^*}(x,a) + \gamma\sigma_{(\mathcal{P}^{s,u^*})_x^a}(V_{\hat{\mathcal{P}}}^{\pi_r})\} \\
&= \max_a\{\hat{r}(x,a) + \gamma\sigma_{\hat{\mathcal{P}}_x^a}(V_{\hat{\mathcal{P}}}^{\pi_r})\} \\
&= V_{\hat{\mathcal{P}}}^{\pi_r}(x),
\end{aligned} \tag{98}$$

which is because $r^{s,u^*}(x,a) = \hat{r}(x,a)$ and $(\mathcal{P}^{s,u^*})_x^a = \hat{\mathcal{P}}_x^a$ for $x \neq s$.

For $s$, we have that

$$\begin{aligned}
\mathbf{T}^{s,u^*}(V_{\hat{\mathcal{P}}}^{\pi_r})(s) &= \max_a\{r^{s,u^*}(s,a) + \gamma\sigma_{(\mathcal{P}^{s,u^*})_s^a}(V_{\hat{\mathcal{P}}}^{\pi_r})\} \\
&= \max_a\{u^* + \gamma\sigma_{(\mathcal{P}^{s,u^*})_s^a}(V_{\hat{\mathcal{P}}}^{\pi_r})\} \\
&= \max_a\{(1-\gamma)V_{\hat{\mathcal{P}}}^{\pi_r}(s) + \gamma(V_{\hat{\mathcal{P}}}^{\pi_r})(s)\} \\
&= V_{\hat{\mathcal{P}}}^{\pi_r}(s),
\end{aligned} \tag{99}$$

which from $(\mathcal{P}^{s,u^*})_s^a = \{\mathbf{1}_s\}$.

Hence combining with eq. (98) implies that $V_{\hat{\mathcal{P}}}^{\pi_r}$ is also a fixed point of $\mathbf{T}^{s,u^*}$, and hence it must be identical to $V^{s,u^*}$, which proves our claim.

**Part 3**. Define a set $U_c \triangleq \left\{ \frac{i}{N} | i = 1, ..., N \right\}$. Clearly, $U_c$ is a $\frac{1}{N}$-net (Vershynin, 2018; Li et al., 2022) of the interval $[0, 1]$.

Note that for any $u \in U_c$, $\mathcal{P}^{s,u}$ is independent with $\hat{\mathsf{P}}_s^a$, hence $V^{s,u}$ is also independent with $\hat{\mathsf{P}}_s^a$. Also, since $u \leq 1$, $\|V^{s,u}\|_\infty \leq \frac{1}{1-\gamma}$.

Then invoking Lemma 5 implies that for any $N(s, a) > 0$, with probability at least $1 - \delta$, it holds simultaneously for all $u \in U_c$ that

$$|(\hat{\mathsf{P}}_s^a - \mathsf{P}_s^a)V^{s,u}| \leq \sqrt{\frac{48\mathbf{Var}_{\hat{\mathsf{P}}_s^a}(V^{s,u}) \log \frac{4N^2}{\delta}}{N(s, a)}} + \frac{48 \log \frac{4N^2}{\delta}}{(1-\gamma)N(s, a)}, \tag{100}$$

$$\mathbf{Var}_{\hat{\mathsf{P}}_s^a}(V^{s,u}) \leq 2\mathbf{Var}_{\mathsf{P}_s^a}(V^{s,u}) + \frac{5 \log \frac{4N^2}{\delta}}{3(1-\gamma)^2 N(s, a)}. \tag{101}$$

**Part 4**. Since $u^* = (1-\gamma)V_{\hat{\mathcal{P}}}^{\pi_r} \leq 1$, then there exists $u_0 \in U_c$, such that $|u_0 - u^*| \leq \frac{1}{N}$. Moreover, we claim that

$$\|V^{s,u^*} - V^{s,u_0}\|_\infty \leq \frac{1}{N(1-\gamma)}. \tag{102}$$

To prove eq. (102), first note that

$$\begin{aligned}
|V^{s,u^*}(s) - V^{s,u_0}(s)| &\leq \max_a |(u^* - u_0) + \gamma(\sigma_{(\mathcal{P}^{s,u^*})_s^a}(V^{s,u^*}) - \sigma_{(\mathcal{P}^{s,u_0})_s^a}(V^{s,u_0}))| \\
&\overset{(a)}{\leq} |u^* - u_0| + \gamma \max_a |\sigma_{(\mathcal{P}^{s,u^*})_s^a}(V^{s,u^*}) - \sigma_{(\mathcal{P}^{s,u^*})_s^a}(V^{s,u_0})| \\
&\overset{(b)}{\leq} |u^* - u_0| + \gamma\|V^{s,u^*} - V^{s,u_0}\|_\infty,
\end{aligned} \tag{103}$$

where $(a)$ is because $(\mathcal{P}^{s,u_0})_s^a = (\mathcal{P}^{s,u^*})_s^a = \{\mathbf{1}_s\}$, and $(b)$ is due to the non-expansion of the support function (Lemma 1, (Panaganti & Kalathil, 2022)).

For $x \neq s$, we have that

$$\begin{aligned}
|V^{s,u^*}(x) - V^{s,u_0}(x)| &\leq \max_a |\hat{r}(x, a) - \hat{r}(x, a) + \gamma(\sigma_{\hat{\mathcal{P}}_x^a}(V^{s,u^*}) - \sigma_{\hat{\mathcal{P}}_x^a}(V^{s,u_0}))| \\
&\leq \gamma\|V^{s,u^*} - V^{s,u_0}\|_\infty.
\end{aligned} \tag{104}$$

Thus by combining eq. (103) and eq. (104), we have that

$$\|V^{s,u^*} - V^{s,u_0}\|_\infty \leq \frac{1}{N} + \gamma\|V^{s,u^*} - V^{s,u_0}\|_\infty, \tag{105}$$

and hence proof the claim eq. (102).

Therefore,

$$\begin{aligned}
&\mathbf{Var}_{\mathsf{P}_s^a}(V^{s,u_0}) - \mathbf{Var}_{\mathsf{P}_s^a}(V^{s,u^*}) \\
&= \mathsf{P}_s^a\left((V^{s,u_0} - \mathsf{P}_s^a V^{s,u_0}) \circ (V^{s,u_0} - \mathsf{P}_s^a V^{s,u_0}) - (V^{s,u^*} - \mathsf{P}_s^a V^{s,u^*}) \circ (V^{s,u^*} - \mathsf{P}_s^a V^{s,u^*})\right) \\
&\overset{(a)}{\leq} \mathsf{P}_s^a\left((V^{s,u_0} - \mathsf{P}_s^a V^{s,u^*}) \circ (V^{s,u_0} - \mathsf{P}_s^a V^{s,u^*}) - (V^{s,u^*} - \mathsf{P}_s^a V^{s,u^*}) \circ (V^{s,u^*} - \mathsf{P}_s^a V^{s,u^*})\right) \\
&\leq \mathsf{P}_s^a\left((V^{s,u_0} - \mathsf{P}_s^a V^{s,u^*} + V^{s,u^*} - \mathsf{P}_s^a V^{s,u^*}) \circ (V^{s,u_0} - V^{s,u^*})\right) \\
&\leq \frac{2}{1-\gamma}|\mathsf{P}_s^a(V^{s,u_0} - V^{s,u^*})|
\end{aligned}$$

$$\leq \frac{2}{N(1-\gamma)^2}, \tag{106}$$

where $(a)$ is due to the fact $\mathbb{E}[X] = \arg\min_c \mathbb{E}[(X-c)^2]$, and the last inequality is due to $\|V^{s,u_0}\|_\infty \leq \frac{1}{1-\gamma}$, $\|V^{s,u^*}\|_\infty \leq \frac{1}{1-\gamma}$.

Similarly, swapping $V^{s,u_0}$ and $V^{s,u^*}$ implies

$$\mathbf{Var}_{\mathsf{P}_s^a}(V^{s,u^*}) - \mathbf{Var}_{\mathsf{P}_s^a}(V^{s,u_0}) \leq \frac{2}{N(1-\gamma)^2}, \tag{107}$$

and further

$$|\mathbf{Var}_{\mathsf{P}_s^a}(V^{s,u^*}) - \mathbf{Var}_{\mathsf{P}_s^a}(V^{s,u_0})| \leq \frac{2}{N(1-\gamma)^2}. \tag{108}$$

We note that eq. (108) is exactly identical to (159) in Section A.4 of (Li et al., 2022), and hence the remaining proof can be obtained by following the proof in Section A.4 in (Li et al., 2022), and is omitted here. $\qquad\square$

**Lemma 7.** *With probability at least $1-\delta$, it holds that for any $s,a$,*

$$\hat{\mathsf{P}}_s^a V_{\hat{\mathcal{P}}}^{\pi_r} - \mathsf{P}_s^a V_{\hat{\mathcal{P}}}^{\pi_r}$$

$$\leq \hat{\mathsf{P}}_s^a V_{\hat{\mathcal{P}}}^{\pi_r} - \sigma_{\hat{\mathcal{P}}_s^a}(V_{\hat{\mathcal{P}}}^{\pi_r}) + 2\sqrt{\frac{48 \log \frac{4SAN}{(1-\gamma)\delta} \epsilon_1}{(1-\gamma)N(s,a)}} + 2\epsilon_1 \sqrt{\frac{48 \log \frac{4SAN}{(1-\gamma)\delta}}{N(s,a)}} + \frac{96 \log \frac{4SAN}{(1-\gamma)\delta}}{(1-\gamma)N(s,a)}. \tag{109}$$

*Proof.* We first show the inequality above holds for any $V \in \left[0, \frac{1}{1-\gamma}\right]$ that is independent with $\hat{\mathsf{P}}_s^a$.

From the duality form of the $\sigma_{\mathcal{P}}(V)$ (Iyengar, 2005), it holds that

$$\hat{\mathsf{P}}_s^a V - \sigma_{\hat{\mathcal{P}}_s^a}(V) = \hat{\mathsf{P}}_s^a V - \max_{\alpha \in [V_{\min}, V_{\max}]} \left\{ \hat{\mathsf{P}}_s^a V_\alpha - \sqrt{R_s^a \mathbf{Var}_{\hat{\mathsf{P}}_s^a}(V_\alpha)} \right\}$$

$$= \min_{\alpha \in [V_{\min}, V_{\max}]} \left\{ \hat{\mathsf{P}}_s^a (V - V_\alpha) + \sqrt{R_s^a \mathbf{Var}_{\hat{\mathsf{P}}_s^a}(V_\alpha)} \right\}, \tag{110}$$

where $V_\alpha \in \mathbb{R}^S$ and $V_\alpha(s) = \min\{V(s), \alpha\}$.

We denote the optimum of the optimization by $\alpha^*$, i.e.,

$$\hat{\mathsf{P}}_s^a (V - V_{\alpha^*}) + \sqrt{R_s^a \mathbf{Var}_{\hat{\mathsf{P}}_s^a}(V_{\alpha^*})} = \min_{\alpha \in [V_{\min}, V_{\max}]} \left\{ \hat{\mathsf{P}}_s^a (V - V_\alpha) + \sqrt{R_s^a \mathbf{Var}_{\hat{\mathsf{P}}_s^a}(V_\alpha)} \right\}. \tag{111}$$

Then eq. (110) can be further bounded as

$$\hat{\mathsf{P}}_s^a V - \sigma_{\hat{\mathcal{P}}_s^a}(V) = \hat{\mathsf{P}}_s^a (V - V_{\alpha^*}) + \sqrt{R_s^a \mathbf{Var}_{\hat{\mathsf{P}}_s^a}(V_{\alpha^*})}$$

$$\geq \hat{\mathsf{P}}_s^a (V - V_{\alpha^*}) - \mathsf{P}_s^a (V - V_{\alpha^*}) + \sqrt{R_s^a \mathbf{Var}_{\hat{\mathsf{P}}_s^a}(V_{\alpha^*})}, \tag{112}$$

where the inequality is due to the fact that $V(s) \geq V_\alpha(s)$.

On the other hand, for any $\alpha \in [V_{\min}, V_{\max}]$ that is fixed and independent with $\hat{\mathsf{P}}_s^a$, we have that

$$\hat{\mathsf{P}}_s^a V - \mathsf{P}_s^a V = \hat{\mathsf{P}}_s^a V_\alpha - \mathsf{P}_s^a V_\alpha + \hat{\mathsf{P}}_s^a (V - V_\alpha) - \mathsf{P}_s^a (V - V_\alpha)$$

$$\leq \hat{\mathsf{P}}_s^a (V - V_\alpha) - \mathsf{P}_s^a (V - V_\alpha) + \frac{48 \log \frac{4SAN}{(1-\gamma)\delta}}{(1-\gamma)N(s,a)} + \sqrt{\frac{48 \log \frac{4SAN}{(1-\gamma)\delta} \mathbf{Var}_{\hat{\mathsf{P}}_s^a}(V_\alpha)}{N(s,a)}}, \tag{113}$$

which is due to $\alpha$ is independent from $\hat{\mathsf{P}}_s^a$, and applying the Bernstein's inequality and (102) of (Shi et al., 2023). Moreover, it implies that

$$\hat{\mathsf{P}}_s^a V - \mathsf{P}_s^a V$$

$$\leq \hat{\mathsf{P}}_s^a(V - V_{\alpha^*}) - \mathsf{P}_s^a(V - V_{\alpha^*}) + \frac{48\log\frac{4SAN}{(1-\gamma)\delta}}{(1-\gamma)N(s,a)} + \sqrt{\frac{48\log\frac{4SAN}{(1-\gamma)\delta}\mathbf{Var}_{\hat{\mathsf{P}}_s^a}(V_{\alpha^*})}{N(s,a)}}$$

$$+ \left(\sqrt{\frac{48\log\frac{4SAN}{(1-\gamma)\delta}\mathbf{Var}_{\hat{\mathsf{P}}_s^a}(V_{\alpha})}{N(s,a)}} - \sqrt{\frac{48\log\frac{4SAN}{(1-\gamma)\delta}\mathbf{Var}_{\hat{\mathsf{P}}_s^a}(V_{\alpha^*})}{N(s,a)}}\right) + (\hat{\mathsf{P}}_s^a(V_{\alpha^*} - V_{\alpha}) - \mathsf{P}_s^a(V_{\alpha^*} - V_{\alpha})). \quad (114)$$

We now construct an $\epsilon_1$-Net (Vershynin, 2018) of $\left[0, \frac{1}{1-\gamma}\right]$ with $\epsilon_1 = \frac{1}{N}$. Specifically, there exists $\mathcal{U} = \left\{\alpha_1, \alpha_2, ..., \alpha_m | \alpha_i \in \left[0, \frac{1}{1-\gamma}\right]\right\}$, such that for any $\alpha \in \left[0, \frac{1}{1-\gamma}\right]$, there exists $\alpha_j \in \mathcal{U}$ with $|\alpha - \alpha_j| \leq \epsilon_1$. Since $\alpha^* \in \left[0, \frac{1}{1-\gamma}\right]$, there exists $\beta \in \mathcal{U}$ with $|\beta - \alpha^*| \leq \epsilon_1$.

It is straightforward to see that

$$\|V_{\alpha^*} - V_{\beta}\| \leq |\beta - \alpha^*| \leq \epsilon_1, \quad (115)$$

and similarly following (207) of (Shi et al., 2023) implies that

$$\left|\sqrt{\mathbf{Var}_{\hat{\mathsf{P}}_s^a}(V_{\beta})} - \sqrt{\mathbf{Var}_{\hat{\mathsf{P}}_s^a}(V_{\alpha^*})}\right| \leq 2\sqrt{\frac{\epsilon_1}{1-\gamma}}. \quad (116)$$

Hence we set $\alpha = \beta$ in eq. (114), take the union bound over $\mathcal{S}, \mathcal{A}$ and $\mathcal{U}$, and plug in the two inequalities above, we have that

$$\hat{\mathsf{P}}_s^a V - \mathsf{P}_s^a V$$

$$\leq \hat{\mathsf{P}}_s^a(V - V_{\alpha^*}) - \mathsf{P}_s^a(V - V_{\alpha^*}) + \frac{48\log\frac{4SAN}{(1-\gamma)\delta}}{(1-\gamma)N(s,a)} + \sqrt{\frac{48\log\frac{4SAN}{(1-\gamma)\delta}\mathbf{Var}_{\hat{\mathsf{P}}_s^a}(V_{\alpha^*})}{N(s,a)}}$$

$$+ 2\sqrt{\frac{48\log\frac{4SAN}{(1-\gamma)\delta}\epsilon_1}{(1-\gamma)N(s,a)}} + 2\epsilon_1\sqrt{\frac{48\log\frac{4SAN}{(1-\gamma)\delta}}{N(s,a)}}. \quad (117)$$

Involving eq. (112) further implies that

$$\hat{\mathsf{P}}_s^a V - \mathsf{P}_s^a V$$

$$\leq \hat{\mathsf{P}}_s^a V - \sigma_{\hat{\mathcal{P}}_s^a}(V) + 2\sqrt{\frac{48\log\frac{4SAN}{(1-\gamma)\delta}\epsilon_1}{(1-\gamma)N(s,a)}} + 2\epsilon_1\sqrt{\frac{48\log\frac{4SAN}{(1-\gamma)\delta}}{N(s,a)}} + \frac{48\log\frac{4SAN}{(1-\gamma)\delta}}{(1-\gamma)N(s,a)}, \quad (118)$$

and hence

$$\sigma_{\hat{\mathcal{P}}_s^a}(V) - \mathsf{P}_s^a V \leq 2\sqrt{\frac{48\log\frac{4SAN}{(1-\gamma)\delta}\epsilon_1}{(1-\gamma)N(s,a)}} + 2\epsilon_1\sqrt{\frac{48\log\frac{4SAN}{(1-\gamma)\delta}}{N(s,a)}} + \frac{48\log\frac{4SAN}{(1-\gamma)\delta}}{(1-\gamma)N(s,a)}. \quad (119)$$

Now we consider $V_{\hat{\mathcal{P}}}^{\pi_r}$. Following the $\frac{1}{N}$-net $\mathcal{U}_2$ we constructed in Lemma 6, $V_{\hat{\mathcal{P}}}^{\pi_r} = V^{s,u}$ for some $u \in [0,1]$. Hence there exists some $u_i \in \mathcal{U}_2$ with $|u - u_i| \leq \frac{1}{N}$. Note that

$$\sigma_{\hat{\mathcal{P}}_s^a}(V_{\hat{\mathcal{P}}}^{\pi_r}) - \mathsf{P}_s^a V_{\hat{\mathcal{P}}}^{\pi_r}$$

$$= \sigma_{\hat{\mathcal{P}}_s^a}(V^{s,u}) - \mathsf{P}_s^a V^{s,u}$$

$$= \sigma_{\hat{\mathcal{P}}_s^a}(V^{s,u_i}) - \mathsf{P}_s^a V^{s,u_i}$$

$$+ \sigma_{\hat{\mathcal{P}}_s^a}(V^{s,u}) - \sigma_{\hat{\mathcal{P}}_s^a}(V^{s,u_i}) + \mathsf{P}_s^a V^{s,u_i} - \mathsf{P}_s^a V^{s,u}$$

$$\leq 2\sqrt{\frac{48\log\frac{4SAN}{(1-\gamma)\delta}\epsilon_1}{(1-\gamma)N(s,a)}} + 2\epsilon_1\sqrt{\frac{48\log\frac{4SAN}{(1-\gamma)\delta}}{N(s,a)}} + \frac{48\log\frac{4SAN}{(1-\gamma)\delta}}{(1-\gamma)N(s,a)}$$
$$+ \sigma_{\hat{\mathcal{P}}_s^a}(V^{s,u}) - \sigma_{\hat{\mathcal{P}}_s^a}(V^{s,u_i}) + \mathsf{P}_s^a V^{s,u_i} - \mathsf{P}_s^a V^{s,u}, \tag{120}$$

which is due to $V^{s,u}$ is independent with $\hat{\mathsf{P}}_s^a$. Moreover, note that both $\sigma_{\hat{\mathcal{P}}_s^a}(V)$ and $\hat{\mathsf{P}}_s^a V$ are 1-Lipschitz, thus

$$\sigma_{\hat{\mathcal{P}}_s^a}(V^{s,u}) - \sigma_{\hat{\mathcal{P}}_s^a}(V^{s,u_i}) \leq \|V^{s,u} - V^{s,u_i}\|_\infty, \tag{121}$$
$$\mathsf{P}_s^a V^{s,u_i} - \mathsf{P}_s^a V^{s,u} \leq \|V^{s,u} - V^{s,u_i}\|_\infty. \tag{122}$$

Now from Equation (102), it follows that

$$\sigma_{\hat{\mathcal{P}}_s^a}(V^{s,u}) - \sigma_{\hat{\mathcal{P}}_s^a}(V^{s,u_i}) \leq \frac{1}{N(1-\gamma)}, \tag{123}$$
$$\mathsf{P}_s^a V^{s,u_i} - \mathsf{P}_s^a V^{s,u} \leq \frac{1}{N(1-\gamma)}. \tag{124}$$

Hence combine with Equation (120), and we have that

$$\sigma_{\hat{\mathcal{P}}_s^a}(V_{\hat{\mathcal{P}}}^{\pi_r}) - \mathsf{P}_s^a V_{\hat{\mathcal{P}}}^{\pi_r}$$
$$\leq 2\sqrt{\frac{48\log\frac{4SAN}{(1-\gamma)\delta}\epsilon_1}{(1-\gamma)N(s,a)}} + 2\epsilon_1\sqrt{\frac{48\log\frac{4SAN}{(1-\gamma)\delta}}{N(s,a)}} + \frac{48\log\frac{4SAN}{(1-\gamma)\delta}}{(1-\gamma)N(s,a)}$$
$$+ \sigma_{\hat{\mathcal{P}}_s^a}(V^{s,u}) - \sigma_{\hat{\mathcal{P}}_s^a}(V^{s,u_i}) + \mathsf{P}_s^a V^{s,u_i} - \mathsf{P}_s^a V^{s,u}$$
$$\leq 2\sqrt{\frac{48\log\frac{4SAN}{(1-\gamma)\delta}\epsilon_1}{(1-\gamma)N(s,a)}} + 2\epsilon_1\sqrt{\frac{48\log\frac{4SAN}{(1-\gamma)\delta}}{N(s,a)}} + \frac{48\log\frac{4SAN}{(1-\gamma)\delta}}{(1-\gamma)N(s,a)} + \frac{2}{N(1-\gamma)}$$
$$\leq 2\sqrt{\frac{48\log\frac{4SAN}{(1-\gamma)\delta}\epsilon_1}{(1-\gamma)N(s,a)}} + 2\epsilon_1\sqrt{\frac{48\log\frac{4SAN}{(1-\gamma)\delta}}{N(s,a)}} + \frac{96\log\frac{4SAN}{(1-\gamma)\delta}}{(1-\gamma)N(s,a)}, \tag{125}$$

which completes the proof. $\qquad\square$

