# OpenReview forum: "Achieving the Asymptotically Minimax Optimal Sample Complexity of Offline Reinforcement Learning: A DRO-Based Approach"
_TMLR — Accepted by TMLR_

### Review · Reviewer_57y6 · 2023-12-14

**Summary Of Contributions:**

This paper proposes a solution to offline reinforcement learning in discounted tabular MDPs,  based on distributionally robust optimization (DRO), and robust value iteration in particular. The proposed algorithm is conceptually simple: use offline data to build an uncertainty set of plausible transition kernels, use it to define a robust MDP, run robust value iteration, and return the optimal robust policy.
This algorithm is shown to achieve similar sample complexity results as LCB-style pessimistic algorithms under a partial coverage assumption. It is also computationally efficient.
A first version, based on rectangular, Hoeffding-style uncertainty sets measured in total-variation, has $O(S^2 C (1-\gamma)^{-4}\epsilon^{-2})$ sample complexity, where S is the number of states and C is the (clipped) single-policy coverage ratio. This is comparable with pessimistic methods based on Hoeffding-style penalties.
A second version, based on Berstein's inequality and chi-square divergence, has $O(SC(1-\gamma)^{-3}\epsilon^{-2})$ sample complexity, matching a minimax lower bound by Rashidinejad et al. (2021) and the best LCB-style algorithms.
The proposed algorithm is tested on two tasks (Garnet and Frozen Lake) in conditions of full and partial coverage, showing performance comparable with LCB.

**Audience:**

Yes

**Broader Impact Concerns:**

Not needed.

**Claims And Evidence:**

Yes

**Requested Changes:**

Critical:
1. The role of the burn-in cost in Theorem 2 should be discussed more, as it seems to defy partial coverage falling back into a uniform-coverage requirement (see the previous section)
2. In the Frozen-Lake environment with partial coverage, LCB improves much faster in the beginning. This deserves at least a comment. Could it be due to the burn-in cost?

Likely just typos:
3. Proof of Thm 1: in the row after Equation (20), in the very last inequality, the $2$ factor seems unnecessary, but actually I think it's the previous inequality that is incorrect.
4. Equation (25), second inequality: this seems off, but likely it's just typos, and the final result seems correct.

Non-critical:
- The proofs could benefit from more comments. For instance, in the proof of Thm3: some intuition could be given on the two-case study (Eq. 17,18); The second inequality of Eq. (22) is a crucial passage and deserves a comment.
- Remark 1 is quite vague. What are the advantages of alternative distance functions? What are the ideas to generalize the proposed method to large-scale problems?
- The second two paragraphs of Section 3 are a needless repetition
- Remark 3: "the motivations are different"...I would rather say that the motivation is the same, but the methods are different. One could even argue that your method is still pessimistic, just with a different implementation of pessimism.
- Experiments: the "partial coverage" setting could be a bit misleading, since full coverage may still be pretty good (every action is played with probability at least $1/(2|A|)$). Of course state coverage heavily depends on the environment, but a comment would be useful here.

Typos etc:
- "transits" page 4
- last equation of page 18: missing plus sign

**Strengths And Weaknesses:**

The main strength of this work is in considering an alternative approach to (provably efficient, tabular) offline RL (robust optimization rather than pessimistic penalties). Equivalently, it could be seen as a retelling or an actualization of robust RL using recent concepts from the offline RL literature. Although this alternative method does not improve over the more common LCB approach, in the spirit of TMLR, it shows an interesting and (I believe) correct alternative path.

Regarding correctness, I checked carefully the proof Theorem 1 (Hoeffding-style uncertainty set) and I did not find any significant errors, although there are some minor issues (likely just typos). A sketch of the Bernstein-style extension is provided in Section 3.2. and it seems correct on a high level. I found the use of the chi-square divergence very interesting from the technical point of view.

My main concern is with the "burn-in" cost from Theorem 2, which scales with the minimum positive probability of the behavior distribution. Although, as correctly observed by the authors, this does not affect in any way the asymptotic convergence, I still think it defies the partial coverage requirement in a way. It looks like it is possible to construct examples where the coverage of the comparator policy is very good, yet the burn-in cost is arbitrarily large. I think this point has been overlooked in the discussion of the result. Moreover, if I am not mistaken, the LCB algorithm by Li et al. does not have this issue, so this raises some doubts on the equivalence of the LCB and the DRO approaches, doubts that are reinforced by the experimental results (see next section).

---

> ### Author Response · Authors · 2024-02-26
>
> We appreciate the thorough review and valuable suggestions provided by the reviewer. In response, we have addressed each comment individually, and the corresponding modifications have been implemented in our paper. The updated parts are highlighted in blue text.
>
> **Burn-in cost and partial coverage assumption.**
> We want to clarify that our result does not transform the partial coverage assumption into a global one. The term $\mu_{\min}$ represents the minimal non-zero entry of the behavior policy $\mu(s,a)$, and our method does not require the dataset to cover all possible state-action pairs.
>
> As noted by the reviewer, $\mu_{\min}$ can lead to a substantial burn-in cost despite the small $C^{\pi^*}$. However, we argue that this doesn't necessarily result in an extreme situation. In fact, the term in the bound should be $\min_s \mu(s,\pi_r(s))$ (which we bounded using $\mu_{\min}$), where $\pi_r$ is the optimal robust policy of the robust MDP we designed. If $\mu(s,a_1)\ll \mu(s,a_2)$ for two actions $a_1,a_2$, then $\pi_r(s)\neq a_1$: if $\mu(s,a_1)$ is too small, the resulting uncertainty set $\mathcal{P}^{a_1}_s$ will be large, leading to a small corresponding value function $Q(s,a_1)<Q(s,a_2)$, making $a_1$ suboptimal. Thus, the term $\min_s \mu(s,\pi_r(s))$ cannot be arbitrarily small since the optimal policy avoids actions that are very rare in the dataset. Hence, our burn-in cost term does not necessarily result in enormous complexity.
>
> However, we acknowledge that it's less strict to assert equivalence between our results and the LCB approach, and claiming minimax optimality may be inaccurate. Therefore, we modify our statement to clarify that our method is **asymptotically** minimax optimal. This means that our result achieves optimal asymptotic complexity (dependent on $\epsilon$), while maintaining a constant-level burn-in cost. Additionally, we emphasize that the primary goal of our work is to address a fundamental question: whether the offline RL problem can be effectively tackled from the perspective of distributional uncertainty, considering limited samples and partial coverage, which has been confirmed by our results.
>
>
>
>
> **The experiment results of Frozen-Lake.**  We thank the reviewer for pointing this out, and we do note that there is a difference in the convergence rate of DRO and LCB approaches. However, we would like to emphasize that in our experiments, $\mu_{\min}=0.5$ and $C^{\pi^*}=2$, resulting in a very small burn-in cost that is overshadowed by the $\epsilon$-dependent term. Therefore, we believe that any observed differences should not be attributed to the burn-in cost but rather to randomness in dataset generation and environmental factors.
>
> To further verify this, we conducted additional experiments in the Frozen-Lake environment with a different behavior action, under both global and partial coverage conditions. The new results are presented in Fig 1 (c,d). As illustrated by both these results and our previous experiments with the garnet problem, there is no conclusive difference between the two methods. We suspect that the burn-in cost would introduce a deterministic difference in our experimental setting, which is not observed.
>
> **Proof of Theorem 1.** We thank the reviewer for pointing the typos out. We have fixed them.

---

> ### Author Response · Authors · 2024-02-26
>
> **Regrading Remark 1: advantages of different functions in designing uncertainty sets, how to generalize to large-scale problems?**
> Firstly, there are no (or at least significant) advantages to using different functions in designing uncertainty sets in the Hoeffding-style construction hence we claim that it is not essential. The core idea is to ensure that the real transition kernel falls into the uncertainty set constructed, and it does not matter how the set is designed. Specifically, we can use different concentration inequalities with different functions, e.g., (Canonne, 2020; Bhandari \& Russo, 2021; Arora et al., 2023) to construct the set, yet all of these constructions will similarly result in sub-optimal complexity, as shown in a concurrent work (Panaganti et al., 2023).
>
> Generalization for large-scale problems is straightforward, especially the Hoeffding-style method. In general, large-scale MDPs may have some low-dimensional structure, e.g., in linear MDP (Jin et al., 2019), the transition kernel $\mathsf{P}^a_{s,s'}=\langle \theta^*(s'), \phi(s,a) \rangle$ can be represented as an inner product of an unknown parameter $\theta^*$ and the pre-set basis functions $\phi$. After a dataset is obtained, we can similarly estimate the parameter $\hat{\theta}$ using, e.g., maximal likelihood estimation, and similarly construct an uncertainty $\mathcal{P}_s=(\{\theta(s): D(\theta(s),\hat{\theta}(s))\leq R_s \})$ such that the true parameter $\theta^*\in \mathcal{P}$. The bound of the sub-optimality can be then similarly derived.
>
> The generalization of the Bernstein-style method is also expected to be doable, yet with more effort. Instead of ensuring $\theta^*\in \mathcal{P}$, we should construct a smaller uncertainty set such that the performance difference between the robust optimal policy and the true policy is small. This design and further analysis, however, require more delicate and careful studies and hence are left for future research.
>
> **Regrading Remark 3.** We agree with the reviewer that the motivations are the same, both to tackle the estimation uncertainty through a pessimistic aspect. The major difference is the method to practically utilize the pessimism principle. The LCB approach obtains a pessimistic estimation of the **value function** by subtracting the penalty term $b$ from the reward. Namely, the result of planning on $M_{LCB}$ is a lower bound of the true value function of the true MDP. In contrast, we would like to answer the question that whether the challenges of limited amount of data and partial coverage in offline RL can be attributed to distributional uncertainty, and whether the problem of offline RL can be solved using distributionally robust optimization approach. The major difference we want to emphasize is that the LCB approach utilizes the pessimistic principle through a conservative **value function** estimation, whereas our DRO approach incorporates the pessimistic principle in designing the **transition models**.
>
>
>
>
> **Partial coverage in the experiment.** Before we generate data, we randomly sample one action $\eta(s)\in\mathcal{A}$ for each state $s$; And during dataset generation, we do not change the action $\eta(s)$ and only obtain samples by taking actions $\pi^*(s)$ or $\eta(s)$. Hence in each run, not all actions are played, but only the optimal one and the pre-set action $\eta(s)$. The dataset hence does not cover all state-action pairs and is a partial coverage one.

---

### Review · Reviewer_h2UY · 2024-02-08

**Summary Of Contributions:**

This paper investigates offline RL with distributionally robust optimization (DRO) based methods. In this robust MDP framework, the transition function is assumed to come from an uncertainty set, and the performance of a policy is measured by its worst performance over all possible transition functions in this set. The goal is to find a policy that obtains the optimal robust performance.

Two sample complexity upper bounds are provided for the discounted tabular MDPs, one with Hoeffding-style uncertainty set and the other with the Bernstein-style uncertainty set. The sharper one with the Bernstein-style uncertainty set matches the minimax lower bound (Rashidinejad et al., 2021) and stands on par with the best results achieved using the LCB approach (Li et al., 2022). The proposed algorithm has a polynomial computational complexity, which is the same as Li et al., 2022 and is better than many other DRO based method that applies more complicated uncertainty set.

Experimental results on Garnet problem and the Frozen-Lake problem are provided to support the theoretical finding.

**Audience:**

Yes

**Claims And Evidence:**

Yes

**Requested Changes:**

Some discussions can be added to make this paper stronger.

In offline RL, it is well known that that there is an equivalence between the value-based method like FQI and the model-based method (using the dataset to construct estimated transition and reward functions and then use the model to do planning like value iteration). The authors remark in the paper that LCB can be viewed as a value-based method where the reward is subtracted by a pessimistic term, while the proposed approach can be viewed as a model-based approach where a transition is first constructed and the planning is conducted (ref eq (10)). Therefore, I was wondering whether a direct equivalence or connection between LCB and the proposed algorithm can be shown.

This paper only assumes that the transition function comes from an uncertainty set. Will the results also hold if the reward function also comes from an uncertainty set?

There’s more related work in offline RL with pessimism:

Chen, Jinglin, and Nan Jiang. "Offline reinforcement learning under value and density-ratio realizability: the power of gaps." In Uncertainty in Artificial Intelligence, pp. 378-388. PMLR, 2022.

**Strengths And Weaknesses:**

**Strength**

The overall presentation and contribution are clear. I believe the major contribution of this paper is it tells readers that DRO-based method can achieve minimax optimal sample complexity bound in offline RL with partial coverage. In addition, the experiment is a plus to this paper.

**Weakness**

The connection between the DRO method and LCB method is unclear (see the section below).

The result only matches the existing (though already minimax optimal) method, and both Hoeffding’s and Bernstein’s bonus/uncertain set construction have been widely investigated in the literature.

Although the rectangular set in DRO yields better computational complexity, I believe it is more restricted than other uncertain set construction.

---

> ### Author Response · Authors · 2024-02-26
>
> We appreciate the thorough review and valuable suggestions provided by the reviewer. In response, we have addressed each comment individually, and the corresponding modifications have been implemented in our paper. The updated parts are highlighted in blue text.
>
>
> **Hoeffding's and Bernstein's bonus/uncertainty set construction have been studied.**
> While we acknowledge that both types of penalty-term constructions have been explored within the framework of the LCB approach, we want to emphasize that our method, which employs these results to form uncertainty sets and subsequently applies DRO to address offline RL, represents a novel contribution to the field, to the best of our knowledge. Although the Hoeffding type designing has been studied in (Uehara \& Sun, 2021), the approach to achieving minimax optimality through the DRO framework and the potential advantages of the Bernstein-style construction, which does not mandate the true transition kernel to reside within the uncertainty set, remain unclear. Our work confirms that despite the possibility of the true kernel not belonging to the set, we can attain superior performance guarantees by adopting a less pessimistic uncertainty set. We believe that our innovative design and theoretical analysis offer a fresh perspective and open up new avenues for addressing offline RL.
>
>
>
>
> **Rectangular sets are more restricted than other constructions.**
> Firstly, we want to clarify that the problem addressed in this paper is a non-robust offline RL problem, meaning that no uncertainty set is inherent to the problem itself. The introduction of an uncertainty set in our method is a deliberate part of the algorithm design, aimed at addressing estimation uncertainty and exploring the feasibility of solving the offline RL problem using robust dynamic programming. As demonstrated theoretically, our adoption of the Bernstein-style uncertainty construction yields a minimax optimal complexity, showcasing its effectiveness.
>
> While it is true that other less-stringent uncertainty sets can be constructed, they often introduce unnecessary challenges and complexities. For instance, constructing $s$-rectangular sets (Wiesemann et al., 2013) ignores the independence of the transition kernels for each state-action pair and introduces unnecessary challenges. Other non-rectangular sets, such as those discussed in (Uehara \& Sun, 2021), face the challenge of NP-hardness and may not be efficiently solvable, and result in a worse sample complexity.
>
> In light of these considerations, we believe and show that the $(s,a)$-rectangular uncertainty set design stands out as the most effective and reasonable construction for addressing offline RL problems, and as such, we have adopted it in our algorithm design.
>
>
> **Connection between LCB and DRO approaches.**
>
> We apologize for any confusion in our paper's description. Both the LCB and our DRO approaches are model-based methods, wherein an MDP model is constructed and planning is performed on it. Our discussion mainly emphasizes the different approaches to tackling estimation error.
>
> The LCB approach constructs an empirical MDP $M_{LCB}=(\mathcal{S},\mathcal{A},\hat{\mathsf{P}},r-b)$ to obtain a pessimistic estimation of the value function by subtracting the penalty term $b$ from the reward. Essentially, planning on $M_{LCB}$ yields a lower bound of the true value function of the MDP. In contrast, our DRO approach does not handle uncertainty through the value function; instead, it constructs statistically plausible sets of transition models to directly account for transition kernel estimation error. The key difference we want to emphasize is that the LCB approach employs the pessimistic principle through a conservative value function estimation, while our DRO approach incorporates the pessimistic principle in directly designing an uncertainty set of transition models, without using the value function as a intermediary step.
>
> Moreover, it's unclear whether one approach could be equivalent to or a special case of the other. While our DRO approach can be reformulated as an LCB approach (as discussed in Remark 3), the penalty term in our approach does not necessarily lead to a conservative estimation of the value function, which renders the results and analysis of LCB approaches invalid.
>
> Regarding the FQI algorithm mentioned by the reviewer, it should indeed be regarded as a model-free method, wherein no model is constructed during learning. However, we leave it as a future interest to design model-free methods for offline RL and explore the connections between them and our DRO methods.

---

> > ### Author Response · Authors · 2024-02-26
> >
> > **Uncertain reward functions.**
> > We assume deterministic reward to streamline our presentation and emphasize our key innovation in addressing uncertainty in offline estimation, particularly in transition kernel estimation, as it presents greater challenges compared to reward estimation.
> >
> > On the other hand,  our methodology is able to be extended for scenarios where reward functions are stochastic, denoted as $r(s,a)\sim \nu$ for some distribution $\nu$ over $\mathcal{R}$. We similarly derive an estimated distribution $\hat{\nu}$ from samples, and construct an uncertainty set $\hat{\mathcal{Q}}^a_s=(\{\tau\in\Delta(\mathcal{R}): D(\tau||\hat{\nu})\leq T^a_s \})$ centered around $\hat{\nu}$. Through a similar design of $T^a_s$ and analysis, we can demonstrate that the worst-case scenario w.r.t. the reward within $\hat{\mathcal{Q}}$ is less than the true expected reward under $\nu$ (up to some small $N^a_s$-dependent terms): $\min_{\tau\in\hat{\mathcal{Q}}^a_s}[r]\leq \mathbb{E}_{\nu}[r]+\mathcal{O}(f(N^a_s))$. This facilitates further analysis similar to the treatment of transition kernel uncertainty, addressing concerns raised by reviewers.
> >
> > It is also noteworthy that formulating robust MDPs to handle both reward and transition dynamic uncertainties can effectively be approached as single-uncertainty robust MDPs, e.g., (Liu et al., 2022) and (Wang et al., 2023), which enables us to adapt our approach to stochastic reward settings seamlessly.
> >
> >
> > **Related work to be added.** We have included a discussion of the suggested work in our paper.

---

### Review · Reviewer_hguE · 2024-02-17

**Summary Of Contributions:**

This paper investigates the sample complexity of offline reinforcement learning from the perspective of Distributionally Robust Optimization (DRO). The central idea involves utilizing empirical model estimation (model-based) as the center of the uncertainty set and appropriately determining the uncertainty radius for transition dynamics. The main contributions are mainly two-fold:
(i) By employing a Hoeffding-style radius, this paper leverages Robust VI (cf. Algorithm 1) with an uncertainty set induced by this radius and establishes a sample complexity of O((1-\gamma)^{-4} \epsilon^{-2} S^2 * C).
(ii) The sample complexity is further enhanced by considering a Bernstein-style uncertainty set, resulting in a sample complexity of O((1-\gamma)^{-3} \epsilon^{-2} S * C), which has been proven to be minimax optimal by [Rashidinejad et al., 2021].
Additionally, the authors validate their algorithms through experimentation on the Garnet Problem.

**Audience:**

Yes

**Broader Impact Concerns:**

There is no Broader Impact Statement section in this paper, and I do not find any immediate concern on the ethical implications of the work .

**Claims And Evidence:**

Yes

**Requested Changes:**

- Based on (8), the corresponding sample complexity in (9) shall also depend on K?
- Is the burn-in cost in [Xie et al., 2021] H^9SC? Should it be 8 in Theorem 4 of [Xie et al., 2021]? (page 9).
- Appendix B, Lemma 2 is the same as Lemma 1? (If yes, please mark them as the same statement.)
- Please define the circle cross notation over sets in page 5.
- What is the norm defined in Lemma 1? (I think it should be L1-norm here) This should be specified in the preliminaries section.
- In Line 11 in Algorithm 2, there should be an “else”? (Otherwise, the pseudo-code will always execute Line 11 to overwrite Line 9 even the condition of Line 8 is satisfied)
- Do the results hold under the less stringent $s$-rectangular setting for the uncertainty set? How much will the sample complexity degenerate when using a general s-rectangular uncertainty set as (Wiesemann et al., 2013)?
- The radius on page 8 should take a min with the Bernstein radius, otherwise the radius is not defined under the (s,a) that satisfies N(s,a) = 0.
- The penalty term in (10) should have a negative sign?

**Strengths And Weaknesses:**

**Strengths**

- This paper establishes that minimax optimal sample complexity can be achieved from the perspective of distributionally robust optimization.
- This work clearly delineates the inherent technical challenges of the direct Hoeffding-style algorithm and addresses them through careful error bounding via utilizing the Bernstein-style uncertainty set.
- They also provide a detailed comparison between their proposed approach and the existing LCB (value-based) approaches, and this strengthens the need for DRO.

**Weaknesses**
- The assumption about deterministic reward functions appears quite strong in this setting. For instance, in the proof of Theorem 2, the authors leverage the fact “r(s,a) >= \hat{r}(s,a)”, however, this does not hold under the random reward function setting, which makes the assumption quite hard to be lifted.
- DRO has been studied in several recent works on offline RL, e.g., (Uehara & Sun, 2021) and (Panaganti et al., 2023), under various assumptions. Similarly, this paper also takes a DRO approach, under somewhat different assumptions. As a result, to make the comparison fair, it is required to also compare the assumptions used in different works (and discuss this in Table 1 as well).
- In Theorem 6 of (Rashidinejad et al., 2021), they introduced a tighter bound (linear dependence of \epsilon) under specific C that lies in [1, 1 + O(lnN / N)]. It would be good to discuss the role of specific C in your algorithm and analyses, and then compare the results with the bound in (Rashidinejad et al., 2021).

---

> ### Author Response · Authors · 2024-02-26
>
> We appreciate the thorough review and valuable suggestions provided by the reviewer. In response, we have addressed each comment individually, and the corresponding modifications have been implemented in our paper. The updated parts are highlighted in blue text.
>
> **W1. The assumption of deterministic reward functions.**
> We adopt this assumption to streamline our presentation and emphasize our key innovation in addressing uncertainty in offline estimation, particularly in transition kernel estimation, as it presents greater challenges compared to reward estimation.
>
> On the other hand,  our methodology is able to be extended for scenarios where reward functions are stochastic, denoted as $r(s,a)\sim \nu$ for some distribution $\nu$ over $\mathcal{R}$. We similarly derive an estimated distribution $\hat{\nu}$ from samples, and construct an uncertainty set  $\hat{\mathcal{Q}}^a_s=( \tau\in\Delta(\mathcal{R}): D(\tau||\hat{\nu})\leq T^a_s )$ centered around $\hat{\nu}$. Through a similar design of $T^a_s$ and analysis, we can demonstrate that the worst-case scenario w.r.t. the reward within $\hat{\mathcal{Q}}$ is less than the true expected reward under $\nu$ (up to some small $N^a_s$-dependent terms): $\min_{\tau\in\hat{\mathcal{Q}}^a_s}[r]\leq \mathbb{E}_{\nu}[r]+\mathcal{O}(f(N^a_s))$. This facilitates further analysis similar to the treatment of transition kernel uncertainty, addressing concerns raised by reviewers.
>
> It is also noteworthy that formulating robust MDPs to handle both reward and transition dynamic uncertainties can effectively be approached as single-uncertainty robust MDPs, e.g., (Liu et al., 2022) and (Wang et al., 2023), which enables us to adapt our approach to stochastic reward settings seamlessly.
>
> **W2. Compare with (Uehara \& Sun, 2021) and (Panaganti et al., 2023), and other offline RL works regarding the assumptions.**
> We would like to clarify that our work, along with the referenced studies, all addresses the offline RL problem within the same partial-coverage setting under the tabular framework. Specifically, these studies assume finite single-policy concentrability: $\max_{s,a}(\frac{d^{\pi^*}(s,a)}{\mu(s,a)})<\infty$, as demonstrated in Corollary 1 of (Uehara \& Sun, 2021) and Theorem 1 of (Panaganti et al., 2023). In our work, we adopt a clipped version of this assumption: $\max_{s,a}(\frac{\min\{d^{\pi^*}(s,a), \frac{1}{S} \}}{\mu(s,a)})< \infty$. It is essential to note that our assumption is weaker than the former, and our results remain valid if we substitute our assumptions with theirs. Therefore, the comparisons outlined in our paper are conducted under the same assumption within the tabular setting, ensuring fairness in the evaluation. We also added the assumption to the comparison table in our paper.
>
> Additionally, we acknowledge that the references mentioned above explore different settings, such as linear MDPs and factored MDPs, which are beyond the scope of our work and thus not included in our comparison.
>
> **W3. Compare with (Rashidinejad et al., 2021) when $C$ falls into a certain region.**
>
> We acknowledge that the LCB approach in (Rashidinejad et al., 2021) achieves lower sample complexity when $C^{\pi^*}$ falls within a specific region. However, we want to point out that there exists a trade-off between complexity results under two scenarios: when $C^{\pi^*}\leq 1+\mathcal{O}(\frac{\log N}{N})$ or $C^{\pi^*}\geq 1+\Omega(\frac{\log N}{N})$.
> The sample complexity depends on the level of pessimism incorporated into algorithm design. The improvement in sample complexity in (Rashidinejad et al., 2021) when $C^{\pi^*}$ is small is at the price of an increased sample complexity of $(1-\gamma)^{-5}$ when $C^{\pi^*}$ is large. Note that our sample complexity $(1-\gamma)^{-4}$ for all $C^{\pi^*}$.
>
> More specifically, when the penalty term is designed large as in (Rashidinejad et al., 2021), the algorithm tends to learn to mimic the most-covered policy by the dataset, akin to imitation learning. When $C^{\pi}$ is small, the most covered policy by dataset is the optimal policy and imitation results in a very small sub-optimality. However, as $C^{\pi}$ increases and the dataset covers more actions, imitation does not necessarily implies optimality, and hence requires more samples. In contrast, our approaches (including both Hoeffding and Bernstein-style approaches) are less conservative and require fewer samples to find the optimal policy under the general case. Yet when $C^{\pi^*}$ is small, our approaches are less conservative (which is harmful in this case) and require more samples to verify whether the behavior policy is optimal, resulting in a larger sample complexity compared to (Rashidinejad et al., 2021). While we attempted to consider different scenarios based on the value of $C^{\pi^*}$, the less conservative nature of our approaches leads to the same complexity under both cases.

---

> > ### Author Response · Authors · 2024-02-26
> > **Requested change part**
> >
> > **Sample complexity in (9) depends on $K$.** The complexity in (9) should indeed be $\mathcal{O}\bigg( {\frac{KSC^{\pi^*}}{(1-\gamma)^3\epsilon^2}}+{\frac{1}{(1-\gamma)KSC^{\pi^*}\mu^2_{\min}}}\bigg)$. However, given that $K$ is a universal constant independent of $S,\gamma,N$, or $C^{\pi^*}$, we omit it from the $\mathcal{O}$-notation.
> >
> > **Burn-in cost in [Xie et al., 2021].** We thank the reviewer for pointing this out. The burn-in cost should be $H^8SC^{\pi^*}$ and has been updated in our paper.
> >
> > **Lemma 2 in Appendix.** Yes they were meant to be the same. We have modified the description in the appendix.
> >
> > **Define $\bigotimes$ and specify the norm.** We apologize for the unclear notations. $\bigotimes$ denotes the Cartesian product of all state-action pairs, indicating that the uncertainty set for each state-action pair is independent of others. For the norm in Lemma 1, it is the $l_1$ norm as we are considering the total variation. We have made them more clear in the paper.
> >
> > **ELSE in  Algorithm 2, there should be an “else”.** We thank you for pointing this out, and there should be an ELSE. If $N(s)>0$ we set the policy at this state as Line 9; And if not, the policy is set as Line 11.
> >
> >
> > **Results for $s$-rectangular setting?**
> >
> > We want to clarify that the choice of uncertainty set structure is due to algorithm design rather than being inherent to the offline RL problem itself, and we have demonstrated that employing $(s,a)$-rectangular uncertainty sets is not only sufficient but also capable of achieving minimax optimality.
> >
> > While it's possible that using $s$-rectangular uncertainty sets could potentially address the offline RL problem, doing so might unnecessarily complicate matters. $s$-rectangular uncertainty sets do not  require independence among uncertainty sets for each state-action pair and are less strict. However, in our specific scenario, where the underlying transition kernels for different state-action pairs are fixed and independent, it's crucial to consider and leverage this independence when designing the uncertainty set. Neglecting this aspect and opting for $s$-rectangular sets could introduce additional challenges and potentially lead to a higher sample complexity.
> >
> > Another drawback of $s$-rectangular sets is their inferior properties compared to $(s,a)$-rectangular sets. For instance, any optimal policy for $s$-rectangular sets may be stochastic (Wiesemann et al., 2013), whereas a deterministic optimal policy always exists for $(s,a)$-rectangular sets. These inferior properties can exacerbate problem tractability issues and yield suboptimal results.
> >
> > Therefore, in addressing whether the DRO framework can effectively and efficiently solve offline RL, we adopt the $(s,a)$-rectangular design and establish its optimality.
> >
> >
> >
> >
> >
> >
> >
> > **Taking min in the Bernstein radius.** When the $N(s,a)=0$, we set the uncertainty set to be the whole probability simplex $\Delta(\mathcal{{S}})$, which means $\chi^2(p||q)\leq \infty$. But we realized the unclear notation here, and have modified it to be more clearly.
> >
> >
> >
> >
> > **The penalty term in (10) should be negative.** We appreciate your comment and have fixed this error.

---

### Decision · Action_Editor_uBJa · 2024-05-31

**Recommendation:** Accept as is

**Comment:**

The reviewers were largely satisfied with this paper, and all their minor concerns were addressed appropriately by the rebuttal. One recurring criticism regarding the contribution has been that the results do not clearly indicate the benefits of the proposed DRO-based approach as compared to the more common LCB-style methods. Despite this limitation, the perspective offered in the paper was deemed interesting, and potentially valuable as a starting point for future research. As all reviewers agreed to publish the paper without further changes, I recommend to go ahead with this decision.

**Audience:**

There is a large community of a researchers working on the theory offline reinforcement learning, who will definitely find the results of this paper relevant and interesting.

**Claims And Evidence:**

The paper is mainly theoretical, and all claims are properly supported by formal proofs.